# Longitudinal proteomic profiling of dialysis patients with COVID-19 reveals markers of severity and predictors of death

Jack Gisby[1†], Candice L Clarke[1,2†], Nicholas Medjeral-Thomas[1,2†], Talat H Malik[1], Artemis Papadaki[1], Paige M Mortimer[1], Norzawani B Buang[1], Shanice Lewis[1], Marie Pereira[1], Frederic Toulza[1], Ester Fagnano[1], Marie-Anne Mawhin[1], Emma E Dutton[1], Lunnathaya Tapeng[1], Arianne C Richard[3,4], Paul DW Kirk[5,6], Jacques Behmoaras[1], Eleanor Sandhu[1,2], Stephen P McAdoo[1,2], Maria F Prendecki[1,2], Matthew C Pickering[1], Marina Botto[1], Michelle Willicombe[1,2‡], David C Thomas[1,2‡], James E Peters[1,7‡*]

[1]Centre for Inflammatory Disease, Department of Immunology and Inflammation, Imperial College London, London, United Kingdom; [2]Renal and Transplant Centre, Hammersmith Hospital, Imperial College Healthcare NHS Trust, London, United Kingdom; [3]Cambridge Institute for Medical Research, University of Cambridge, Cambridge, United Kingdom; [4]CRUK Cambridge Institute, University of Cambridge, Cambridge, United Kingdom; [5]MRC Biostatistics Unit, Forvie Way, University of Cambridge, Cambridge, United Kingdom; [6]Cambridge Institute of Therapeutic Immunology & Infectious Disease, University of Cambridge, Cambridge, United Kingdom; [7]Health Data Research UK, London, United Kingdom

**\*For correspondence:**
j.peters@imperial.ac.uk

[†]These authors contributed equally to this work
[‡]These authors also contributed equally to this work

**Abstract** End-stage kidney disease (ESKD) patients are at high risk of severe COVID-19. We measured 436 circulating proteins in serial blood samples from hospitalised and non-hospitalised ESKD patients with COVID-19 (n = 256 samples from 55 patients). Comparison to 51 non-infected patients revealed 221 differentially expressed proteins, with consistent results in a separate subcohort of 46 COVID-19 patients. Two hundred and three proteins were associated with clinical severity, including IL6, markers of monocyte recruitment (e.g. CCL2, CCL7), neutrophil activation (e.g. proteinase-3), and epithelial injury (e.g. KRT19). Machine-learning identified predictors of severity including IL18BP, CTSD, GDF15, and KRT19. Survival analysis with joint models revealed 69 predictors of death. Longitudinal modelling with linear mixed models uncovered 32 proteins displaying different temporal profiles in severe versus non-severe disease, including integrins and adhesion molecules. These data implicate epithelial damage, innate immune activation, and leucocyte–endothelial interactions in the pathology of severe COVID-19 and provide a resource for identifying drug targets.

## Introduction

Coronavirus disease (COVID-19), caused by the SARS-CoV-2 virus, displays wide clinical heterogeneity from asymptomatic to fatal disease. Patients with severe disease exhibit marked inflammatory responses and immunopathology. The mechanisms underlying this remain incompletely characterised, and the key molecular mediators are yet to be determined. The first treatment shown to reduce mortality from COVID-19 in randomised trials was dexamethasone (*Horby et al., 2020*), a

**eLife digest** COVID-19 varies from a mild illness in some people to fatal disease in others. Patients with severe disease tend to be older and have underlying medical problems. People with kidney failure have a particularly high risk of developing severe or fatal COVID-19.

Patients with severe COVID-19 have high levels of inflammation, causing damage to tissues around the body. Many drugs that target inflammation have already been developed for other diseases. Therefore, to repurpose existing drugs or design new treatments, it is important to determine which proteins drive inflammation in COVID-19.

Here, Gisby, Clarke, Medjeral-Thomas et al. measured 436 proteins in the blood of patients with kidney failure and compared the levels between patients who had COVID-19 to those who did not. This revealed that patients with COVID-19 had increased levels of hundreds of proteins involved in inflammation and tissue injury. Using a combination of statistical and machine learning analyses, Gisby et al. probed the data for proteins that might predict a more severe disease progression. In total, over 200 proteins were linked to disease severity, and 69 with increased risk of death. Tracking how levels of blood proteins changed over time revealed further differences between mild and severe disease. Comparing this data with a similar study of COVID-19 in people without kidney failure showed many similarities. This suggests that the findings may apply to COVID-19 patients more generally.

Identifying the proteins that are a cause of severe COVID-19 – rather than just correlated with it – is an important next step that could help to select new drugs for severe COVID-19.

corticosteroid that has broad non-specific effects on the immune system. Even with corticosteroid treatment, mortality in severe COVID-19 remains significant. There is a wide armamentarium of existing drugs that target inflammation more selectively, providing potential repurposing opportunities for the treatment of COVID-19. Recently, the REMAP-CAP trial has demonstrated efficacy of anti-IL6 receptor blockade in patients admitted to intensive care units with severe disease (*Gordon et al., 2021*). In order to select the most promising agents for future trials, we urgently need to better understand the molecular drivers of severe disease. Proteins are the effector molecules of biology and the targets of most drugs. Therefore, proteomic profiling to identify the key mediators of severe disease provides a valuable tool for identifying and prioritising potential drug targets (*Suhre et al., 2021*).

Risk factors for severe or fatal COVID-19 include age, male sex, non-European ancestry, obesity, diabetes mellitus, cardiovascular disease, and immunosuppression (*Williamson et al., 2020*). End-stage kidney disease (ESKD) is one of the strongest risk factors for severe COVID-19 (estimated hazard ratio for death 3.69) (*Williamson et al., 2020*), and ESKD patients hospitalised with COVID-19 have a mortality of approximately 30% (*Docherty et al., 2020*; *Corbett et al., 2020*; *Ng et al., 2020*; *Valeri et al., 2020*). ESKD patients have a high prevalence of vascular and cardiometabolic disease (e.g. hypertension, ischaemic heart disease, diabetes), either as a result of the underlying cause of their renal disease or as a consequence of renal failure. In addition, ESKD results in both relative immunosuppression and chronic low-grade inflammation, which may impact viral defence and the host inflammatory response.

Here we performed proteomic profiling of serial blood samples of ESKD patients with COVID-19, leveraging the unique opportunity for longitudinal sampling in both the outpatient and inpatient settings afforded by a large multi-ethnic haemodialysis cohort (*Figure 1a*). These data revealed 221 proteins that are dysregulated in COVID-19 versus matched non-infected ESKD patients. Using linear mixed models, joint models, and machine learning, we identified proteins that are markers of COVID-19 severity and risk of death. Finally, we characterised the temporal dynamics of the blood proteomic response during COVID-19 infection in ESKD patients, uncovering 32 proteins that display altered trajectories in patients with severe versus non-severe disease.

## Results

We recruited 55 ESKD patients with COVID-19 (subcohort A; *Table 1*). All patients were receiving haemodialysis prior to acquiring COVID-19. Blood samples were taken as soon as feasible following

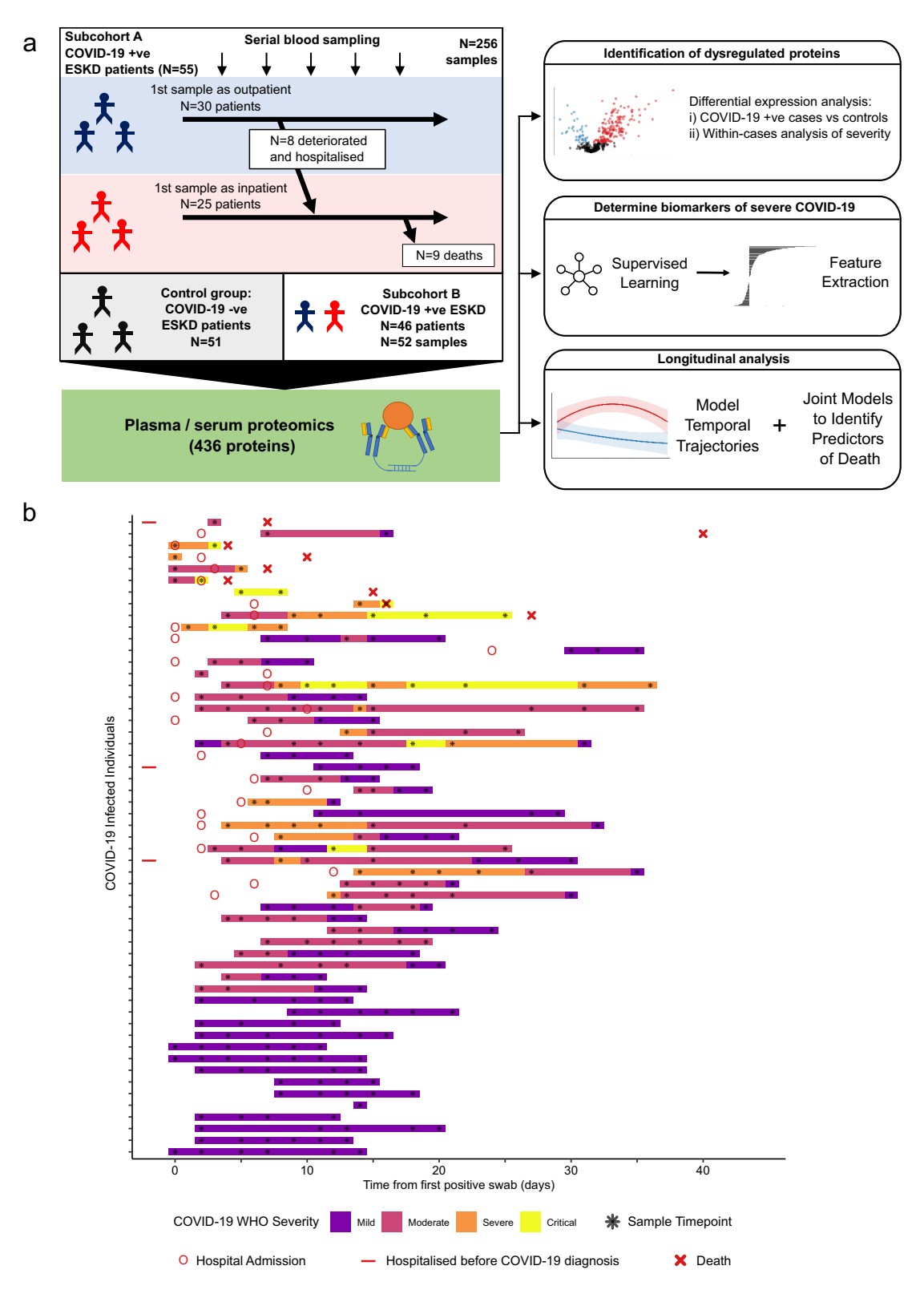

**Figure 1.** Study design. (a) Schematic representing a summary of the patient cohorts, sampling, and the major analyses. Blue and red stick figures represent outpatients and hospitalised patients, respectively. (b) Timing of serial blood sampling in relation to clinical course of COVID-19 (subcohort A). Black asterisks indicate when samples were obtained. Three patients were already in hospital prior to COVID-19 diagnosis (indicated by red bars).

*Figure 1 continued on next page*

*Figure 1 continued*

The online version of this article includes the following figure supplement(s) for figure 1:

**Figure supplement 1.** Baseline characteristics of subcohort A.

COVID-19 diagnosis. At time of initial sample, 30 patients were outpatients attending haemodialysis sessions and 25 were hospitalised inpatients (see Materials and methods, *Figure 1*). Following the initial blood sample, serial sampling was performed for 51/55 patients. We also recruited 51 non-infected haemodialysis patients as ESKD controls, mirroring the age, sex, and ethnicity distribution of the COVID-19 cases (*Figure 1—figure supplement 1a–c*). We used the Olink proteomics platform to measure 436 proteins (*Supplementary file 1a*) in 256 plasma samples from the COVID-19 patients and the 51 control samples. The proteins measured consisted of five multiplex 'panels' focussed on proteins relevant to immuno-inflammation, cardiovascular, and cardiometabolic disease. The 436 proteins assayed showed strong enrichment for immune-related proteins (*Supplementary file 1b*).

In addition, we performed the Olink proteomic assays in 52 serum samples from a separate set of 46 COVID-19-positive ESKD patients (subcohort B) and 11 serum samples from ESKD COVID-19-negative controls (a subset of the controls described above). For the large majority of patients in subcohort B, only a single timepoint was available. A higher proportion of these patients (41/46, 89%) were hospitalised and had severe disease (*Table 2*) than in subcohort A (*Figure 1*, *Table 1*).

**Table 1.** Characteristics of subcohort A.

| | COVID-19-positive ESKD patients (n = 55) | | | ESKD controls (n = 51) |
|---|---|---|---|---|
| | Overall | Peak severity mild or moderate (n = 28) | Peak severity severe or critical (n = 27) | |
| **Age** | | | | |
| Median | 72.2 | 73.4 | 68.5 | 70.1 |
| (IQR) | 62.5–77.3 | 65.5–76.4 | 61.8–78.8 | 62.2–75.1 |
| **Sex** | | | | |
| M | 39 (70.9%) | 18 (64.3%) | 21 (77.8%) | 36 (70.6%) |
| F | 16 (29.1%) | 10 (35.7%) | 6 (22.2%) | 15 (29.4%) |
| **Ethnicity** | | | | |
| White | 16 (29.1%) | 5 (17.9%) | 11 (40.7%) | 13 (25.5%) |
| Black | 8 (14.5%) | 5 (17.9%) | 3 (11.1%) | 8 (15.7%) |
| South Asian | 18 (32.7%) | 10 (35.7%) | 8 (29.6%) | 20 (39.2%) |
| Asian (other) | 4 (7.3%) | 1 (3.6%) | 3 (11.1%) | 3 (5.9%) |
| Other | 9 (16.4%) | 7 (25.0%) | 2 (7.4%) | 7 (13.7%) |
| Diabetes | 34 (61.8%)[*] | 16 (57.1%) | 18 (66.7%) | 24 (47.1%)[*] |
| Current smoker | 1 (1.8%) | 1 (3.6%) | 0 | 0 |
| **ESKD cause** | | | | |
| DN | 29 (52.7%) | 14 (50.0%) | 15 (55.6%) | 20 (39.2%) |
| Genetic | 1 (1.8%) | 1 (3.6%) | 0 | 1 (2.0%) |
| GN | 3 (5.5%) | 1 (3.6%) | 2 (7.4%) | 9 (17.6%) |
| HTN/vascular | 5 (9.1%) | 3 (10.7%) | 2 (7.4%) | 7 (13.7%) |
| Other | 8 (14.5%) | 5 (17.9%) | 3 (11.1%) | 4 (7.8%) |
| Unknown | 9 (16.4%) | 4 (14.3%) | 5 (18.5%) | 10 (19.6%) |
| Hospitalisation due to COVID-19† | 33 (60%) | 6 (21.4%) | 27 (100%) | N/A |
| Fatal COVID-19 | 9 (16.3%) | 0 (0%) | 9 (33.3%) | N/A |

DN = diabetic nephropathy. GN = glomerulonephritis. HTN = hypertension. IQR = inter-quartile range. 'South Asian' represents individuals with Indian, Pakistani, or Bangladeshi ancestry. Subsets defined according to peak WHO severity over the course of the illness. N/A = not applicable.

[*]One patient had type 1 diabetes, the remainder type 2. †3 patients were hospitalised prior to COVID-19 diagnosis. 8 patients diagnosed with COVID-19 as outpatients subsequently deteriorated were hospitalised.

**Table 2.** Characteristics of subcohort B.

| | COVID-19-positive ESKD patients (n = 46) | COVID-19-negative ESKD controls (n = 11)* |
|---|---|---|
| **Age** | | |
| Median | 64.3 | 71.6 |
| (IQR) | 60.3–73.0 | (61.7–73.9) |
| **Sex** | | |
| M | 32 (69.6%) | 8 (72.3%) |
| F | 14 (30.4%) | 3 (27.3%) |
| **Ethnicity** | | |
| White | 11 (23.9%) | 3 (27.3%) |
| Black | 8 (17.4%) | 3 (27.3%) |
| South Asian | 12 (26.1%) | 3 (27.3%) |
| Asian (other) | 7 (15.2%) | 0 |
| Other | 8 (17.4%) | 2 (18.2%) |
| **Diabetes** | 29 (63.0%) | 6 (54.5%) |
| **Current smoker** | 2 (4.3%) | 0 (%) |
| **ESKD cause** | | |
| DN | 19 (41.3%) | 5 (45.5%) |
| Genetic | 1 (2.2%) | 0 |
| GN | 7 (15.2%) | 1 (9.1%) |
| HTN/vascular | 3 (6.5%) | 1 (9.1%) |
| Other | 3 (6.5%) | 2 (18.2%) |
| Unknown | 13 (28.3%) | 2 (18.2%) |
| **Hospitalisation due to COVID-19** | 41 (89.1%) | N/A |
| **Severe or critical COVID-19** | 33 (71.7%) | N/A |
| **Fatal COVID-19** | 9 (19.6%) | N/A |

DN = diabetic nephropathy. GN = glomerulonephritis. HTN = hypertension. IQR = inter-quartile range. 'South Asian' represents individuals with Indian, Pakistani, or Bangladeshi ancestry. Subsets defined according to peak WHO severity over the course of the illness. N/A = not applicable. *These 11 controls are a subset of the control patients used in subcohort A.

## Proteomic differences between COVID-19-positive and -negative ESKD patients

Principal component analysis (PCA) of proteomic data from subcohort A demonstrated differences between samples from COVID-19-positive cases and controls, although the two groups did not separate into discrete clusters (*Figure 2a,b*). To examine the effects of COVID-19 on the plasma proteome, we performed a differential expression analysis in subcohort A between COVID-19 cases (n = 256 samples passing quality control [QC] from 55 patients) and non-infected ESKD controls (n = 51) using linear mixed models, which account for serial samples from the same individual (see Materials and methods). This revealed 221 proteins associated with COVID-19 (5% false discovery rate, FDR); the vast majority were upregulated, with only 40 downregulated (*Figure 3a*, *Supplementary file 1c*). In order to provide a succinct and standardised nomenclature, we report proteins by the symbols of the genes encoding them (see *Supplementary file 1a* for a mapping of symbols to full protein names). The most strongly upregulated proteins (in terms of fold change) were DDX58, CCL7, IL6, CXCL11, KRT19, and CXCL10, and the most strongly downregulated were SERPINA5, CCL16, FABP2, PON3, ITGA11, and MMP12 (*Figure 3—figure supplement 1*). Notably, many of the upregulated proteins were chemotaxins.

We observed that a high proportion of the measured proteins were associated with COVID-19. Given the highly targeted nature of the Olink panels that we used (enriched for immune and inflammation-related proteins), this was not surprising. Nevertheless, to ensure that the Benjamini–Hochberg adjustment of p-values was controlling the FDR at the 5% level, we performed two additional analyses (see Materials and methods). First, we estimated the FDR using an alternative method (the plug-in procedure ; *Hastie et al., 2001*); this confirmed appropriate FDR control. Second, we used permutation to estimate the distribution of the number of proteins expected to be declared significant under the null hypothesis of no association between any proteins and COVID-19. This showed

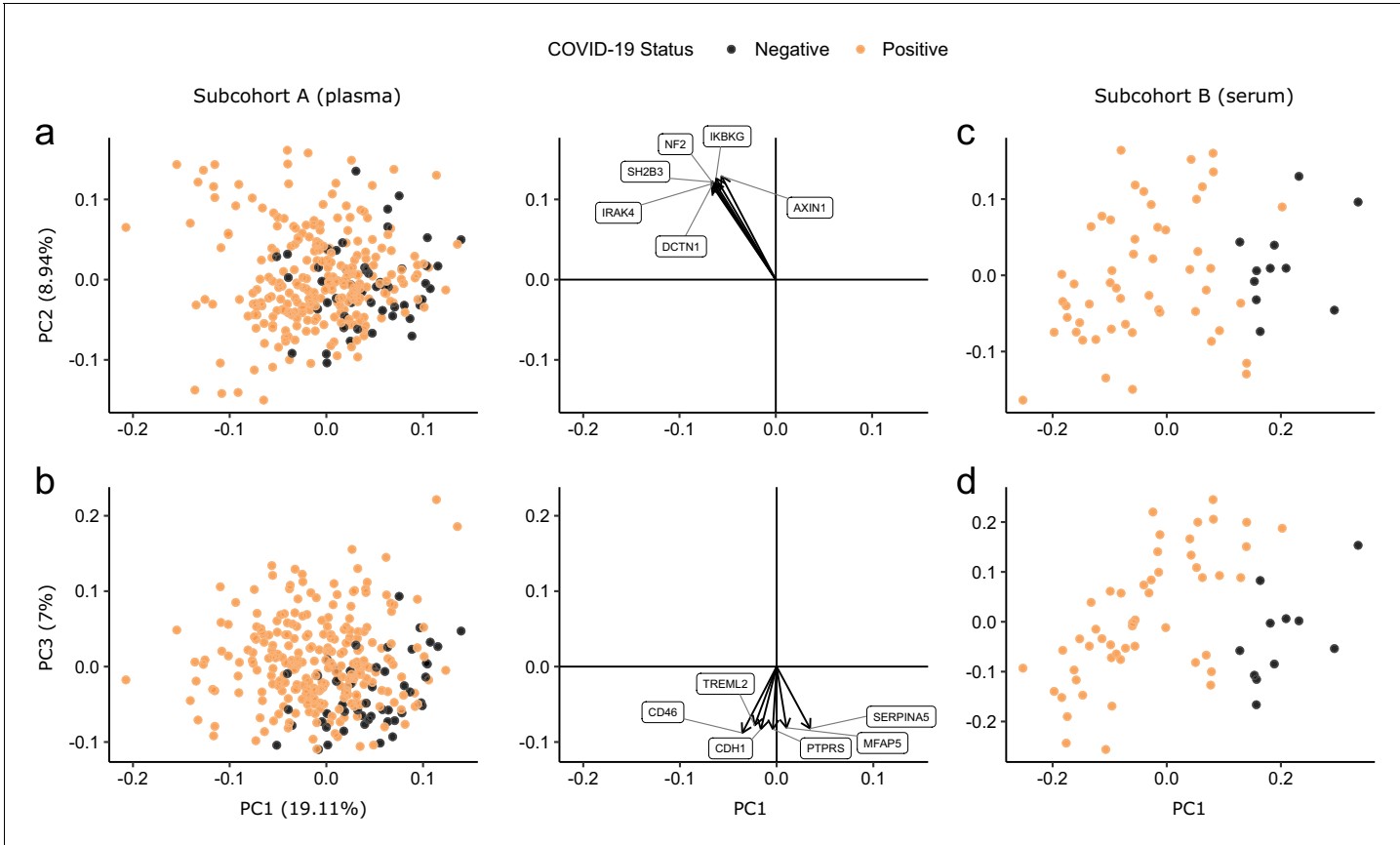

**Figure 2.** Principal component analysis. PC = principal component. Each point represents a sample. Colouring indicates COVID-19 status. The directions and relative sizes of the six largest PC loadings are plotted as arrows (middle column). (a, b) Subcohort A. Due to serial sampling, there are multiple samples for most patients. The proportion of variance explained in subcohort A by each PC is shown in parentheses on the axis labels. (c, d) Subcohort B. Samples are projected into the PCA coordinates from subcohort A.

The online version of this article includes the following figure supplement(s) for figure 2:

**Figure supplement 1.** Principal component analysis in relation to clinical severity.
**Figure supplement 2.** Principal component analysis in relation to assay plate.

that the probability of observing the number of differentially abundant proteins we identified was highly unlikely under the null (empirical $p<1\times10^{-5}$; *Figure 3—figure supplement 2*).

Although our COVID-19-negative controls were well matched in terms of age, sex, and ethnicity (*Figure 1—figure supplement 1a–c*), perfect matching of comorbidities was not feasible in the context of the healthcare emergency at the time of patient recruitment. There was a higher prevalence of diabetes in the COVID-19 cases compared to the controls (61.8% versus 47.1%, respectively; *Table 1*). To evaluate whether differing rates of diabetes had impacted the proteins identified as differentially abundant between cases and controls, we performed a sensitivity analysis adding diabetes as an additional covariate in the linear mixed model. This did not materially affect our findings; estimated effect sizes and –log10 p-values from models with and without the inclusion of diabetes were highly correlated (Pearson r > 0.99, and r = 0.95, respectively; *Figure 3—figure supplement 3a,b*). Full results from both models are shown in *Supplementary file 1c*. Similarly, there were also differences in the underlying cause of ESKD in cases compared to controls (*Table 1*). We therefore performed a further sensitivity analysis adjusting for underlying cause of renal failure. This did not make any meaningful difference to our results (*Figure 3—figure supplement 3c,d*, *Supplementary file 1c*).

We also considered the possibility that timing of haemodialysis might affect the plasma proteome. To minimise the impact of this, all samples were taken prior to haemodialysis. For the large

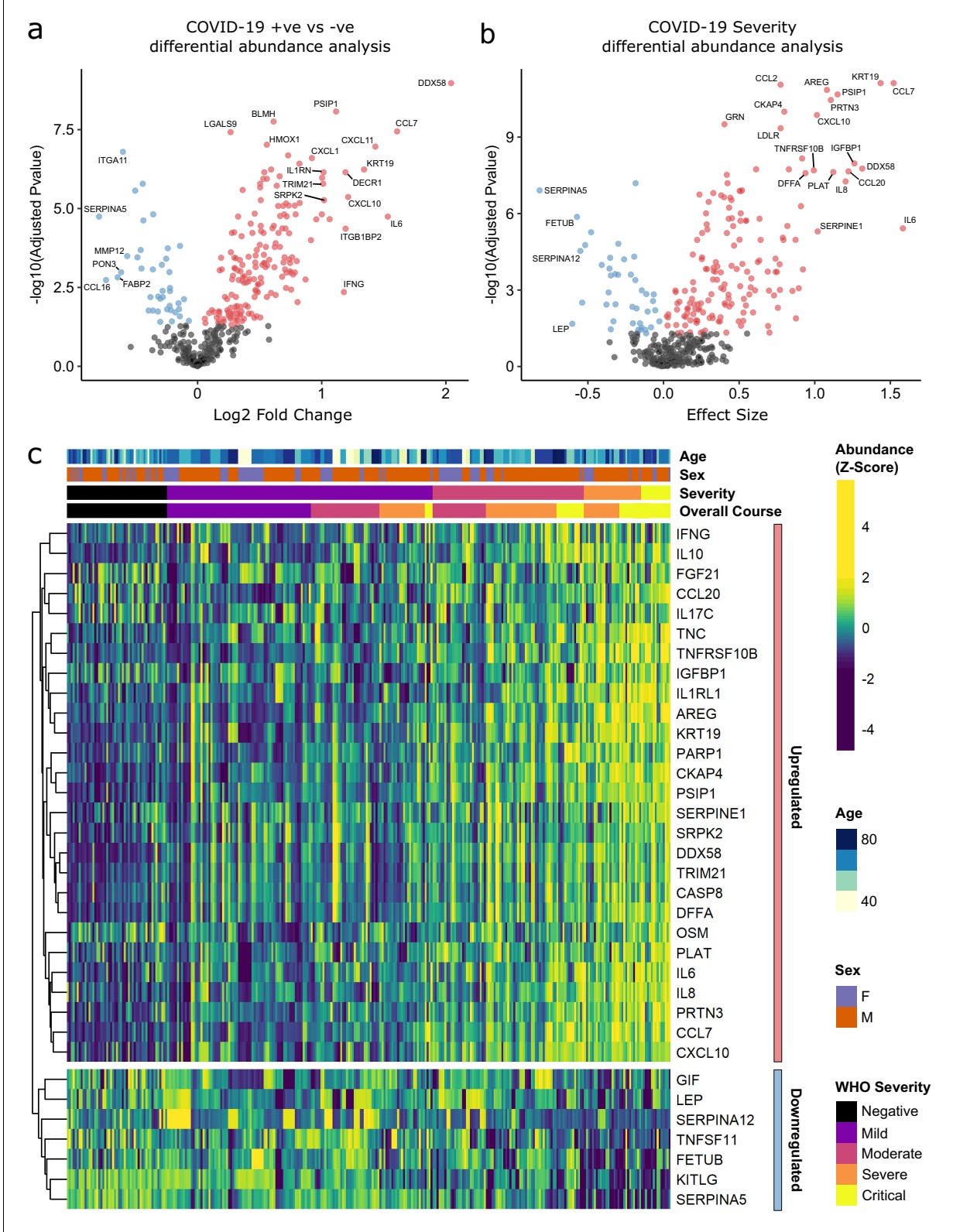

**Figure 3.** Identification of dysregulated proteins. (**a**) Proteins upregulated (red) or downregulated (blue) in COVID-19-positive patients versus COVID-19-negative ESKD patients n = 256 plasma samples from 55 COVID-19-positive patients, versus n = 51 ESKD controls (one sample per control patient). (**b**) Proteins associated with disease severity associations of protein levels against WHO severity score at the time of sampling. Linear gradient indicates the effect size. A positive effect size (red) indicates that an increase in protein level is associated with increasing disease severity and a negative

*Figure 3 continued on next page*

*Figure 3 continued*

gradient (blue) the opposite. n = 256 plasma samples from 55 COVID-19-positive patients. For (**a**, **b**), p-values from linear mixed models after Benjamini–Hochberg adjustment; significance threshold = 5% FDR; dark-grey = non-significant. (**c**) Heatmap showing protein levels for selected proteins with strong associations with severity. Each column represents a sample (n = 256 COVID-19 samples and 51 non-infected samples). Each row represents a protein. Proteins are annotated using the symbol of their encoding gene. For the purposes of legibility, not all significantly associated proteins are shown; the heatmap is limited to the 17% most up- or downregulated proteins (by effect size) of those with a significant association. Proteins are ordered by hierarchical clustering. Samples are ordered by WHO severity at the time of blood sample ('Severity'). 'Overall course' indicates the peak WHO severity over the course of the illness.

The online version of this article includes the following figure supplement(s) for figure 3:

**Figure supplement 1.** Differential abundance analysis between ESKD patients with and without COVID-19.

**Figure supplement 2.** Permutation analysis to estimate the null distribution.

**Figure supplement 3.** Sensitivity analyses adjusting for diabetes status and cause of ESKD.

**Figure supplement 4.** Sensitivity analysis adjusting for time since last haemodialysis.

---

majority (86.6%) of samples, the most recent haemodialysis was between 48 and 72 hr prior to blood draw. This consistency in timing of blood sampling reduces the potential for impact of this issue. Nevertheless, to evaluate whether timing of haemodialysis might have impacted our results, we performed a sensitivity analysis including time from last haemodialysis as a covariate. Our results were not materially affected by this, with −log10 p-values and estimated effect sizes very highly correlated with those obtained without inclusion of this covariate (Pearson r > 0.99 for effect size estimates and for −log10 p-values; *Figure 3—figure supplement 4a,b*, *Supplementary file 1c*).

We used the smaller subcohort B (n = 52 serum samples from 46 patients with COVID-19; see Materials and methods) for validation. We first projected the data from subcohort B into the PCA space of subcohort A to examine how well the separation of cases and controls in the PCA space replicated (see Materials and methods). This revealed clearer separation of infected and non-infected patients than in subcohort A (*Figure 2c,d*), perhaps reflecting the higher proportion of hospitalised patients (41 of 46 patients) in subcohort B (*Table 2*). We next performed differential abundance analysis in subcohort B and found 201 proteins that were dysregulated in cases versus controls (5% FDR) (*Supplementary file 1c*). Of the 221 differentially abundant proteins from subcohort A, 150 (69.7%) were also identified in subcohort B at 5% FDR (*Figure 4a*). Effect sizes in each dataset showed a strong correlation (r = 0.80, *Figure 4b*). This demonstrates that our findings are highly reproducible despite differences in sample sizes and blood materials (plasma versus serum in subcohorts A and B, respectively).

## Proteins associated with COVID-19 severity

Examination of the principal components plot labelling samples by clinical severity at the time of sampling (defined by WHO severity scores, graded as mild, moderate, severe, or critical) demonstrated a gradient of COVID-19 severity, best captured by principal components 1 and 3 (*Figure 2—figure supplement 1a*). To determine the proteomic effects of COVID-19 severity, we tested for associations between proteins and WHO severity score at the time of blood sampling, using linear mixed models with severity encoded as an ordinal predictor (see Materials and methods). This analysis revealed 203 proteins associated with severity (*Figure 3b*, *Supplementary file 1d*). The majority of these were upregulated in more severe disease, with only 42 downregulated. A sensitivity analysis adjusting for time since last haemodialysis made no significant impact on our results (*Figure 3—figure supplement 4c,d*, *Supplementary file 1d*). Consistent with previous reports, we found that severe COVID-19 was characterised by elevated IL6. In addition, we observed a signature of upregulated monocyte chemokines (e.g. CCL2, CCL7, CXCL10), neutrophil activation and degranulation (e.g. PRTN3, MPO), and epithelial injury (e.g. KRT19, AREG, PSIP1, GRN). (*Figures 3b,c* and *5*). SERPINA5 and leptin showed the greatest downregulation as COVID-19 severity increased (*Figure 3b,c*).

We next asked how does the COVID-19 severity protein signature relate to the proteins that are differentially abundant between cases and controls? The majority (140/203; 69%) of severity-associated proteins were also identified as differentially abundant in the COVID-19-positive versus -negative analysis (*Figure 6a*). Log fold changes for proteins in COVID-19 versus non-infected patients were correlated with effect sizes in the severity analysis, such that the proteins most upregulated in

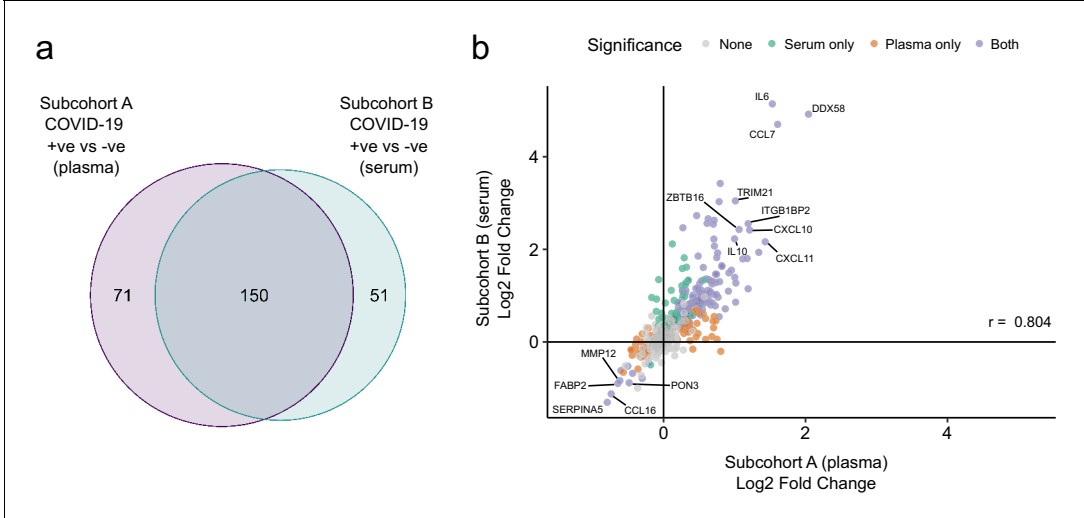

**Figure 4.** Validation. (a) Overlap between the significant associations in the differential abundance analysis between ESKD patients with and without COVID-19 in subcohorts A and B. 5% FDR was used as the significance threshold in both analyses. (b) Comparison of estimated effect sizes for all 436 proteins in the differential abundance analyses (COVID-19 positive versus negative) in subcohort A and B. Each point represents a protein. Pearson's r is shown. Differential abundance analyses were performed using linear mixed models. Subcohort A analysis (plasma samples): 256 samples from 55 COVID-19 patients versus 51 non-infected patient samples (single time-point). Subcohort B (serum samples): 52 samples from 55 COVID-19 patients and 11 non-infected patient samples (single timepoint).

The online version of this article includes the following figure supplement(s) for figure 4:

**Figure supplement 1.** Comparison with the report of *Filbin et al., 2020*.

cases versus controls also tended to show the greatest upregulation in severe disease (*Figure 6b*). However, there were some notable exceptions (e.g. CCL20, IL17C, OSM) that were strongly associated with severity, but not differentially expressed in infected versus non-infected patients (*Figure 6c*).

## Supervised learning to predict COVID-19 severity

PCA revealed that some samples from patients who had mild or moderate disease at the time of sampling clustered with samples from patients with severe disease (*Figure 2—figure supplement 1a*). Examination of the same PCA plot labelling samples according to the patient's overall clinical course (measured by peak WHO severity score over the duration of the illness) (*Figure 2—figure supplement 1b*) revealed that these samples came from individuals who subsequently developed severe or critical disease. This suggested that molecular changes may predate clinical deterioration. To evaluate this further, we used supervised learning approaches to test whether the proteomic signature of the first blood sample for each patient in our dataset could identify whether the patient either had clinically severe COVID-19 at the time of sampling or would develop severe disease in the future. Whereas differential expression analyses consider each protein marker separately, machine-learning techniques allow examination of all proteins concurrently, thus capturing non-linear relationships in the dataset. Using Random Forests, we trained a classifier on the first sample for each COVID-19 patient to predict the overall clinical course, defined by peak WHO severity. For the purposes of this analysis, we binarised clinical course into either WHO mild/moderate or severe/critical.

The Random Forests method achieved 71% accuracy in predicting peak severity. By contrast, using only clinically available predictors (demographics, comorbidities, and clinical laboratory results), the Random Forests method achieved 66% accuracy in predicting peak severity. Combining clinical parameters plus proteins did not improve accuracy (71%) compared to using proteomic predictors alone, suggesting that the information contained in the clinical predictors is captured at the proteomic level. While we do not believe that proteomic profiling is likely to enter clinical practice for risk stratification during this pandemic, the features selected by the classifier can highlight proteins of biological importance. We therefore interrogated the model to identify key proteins by

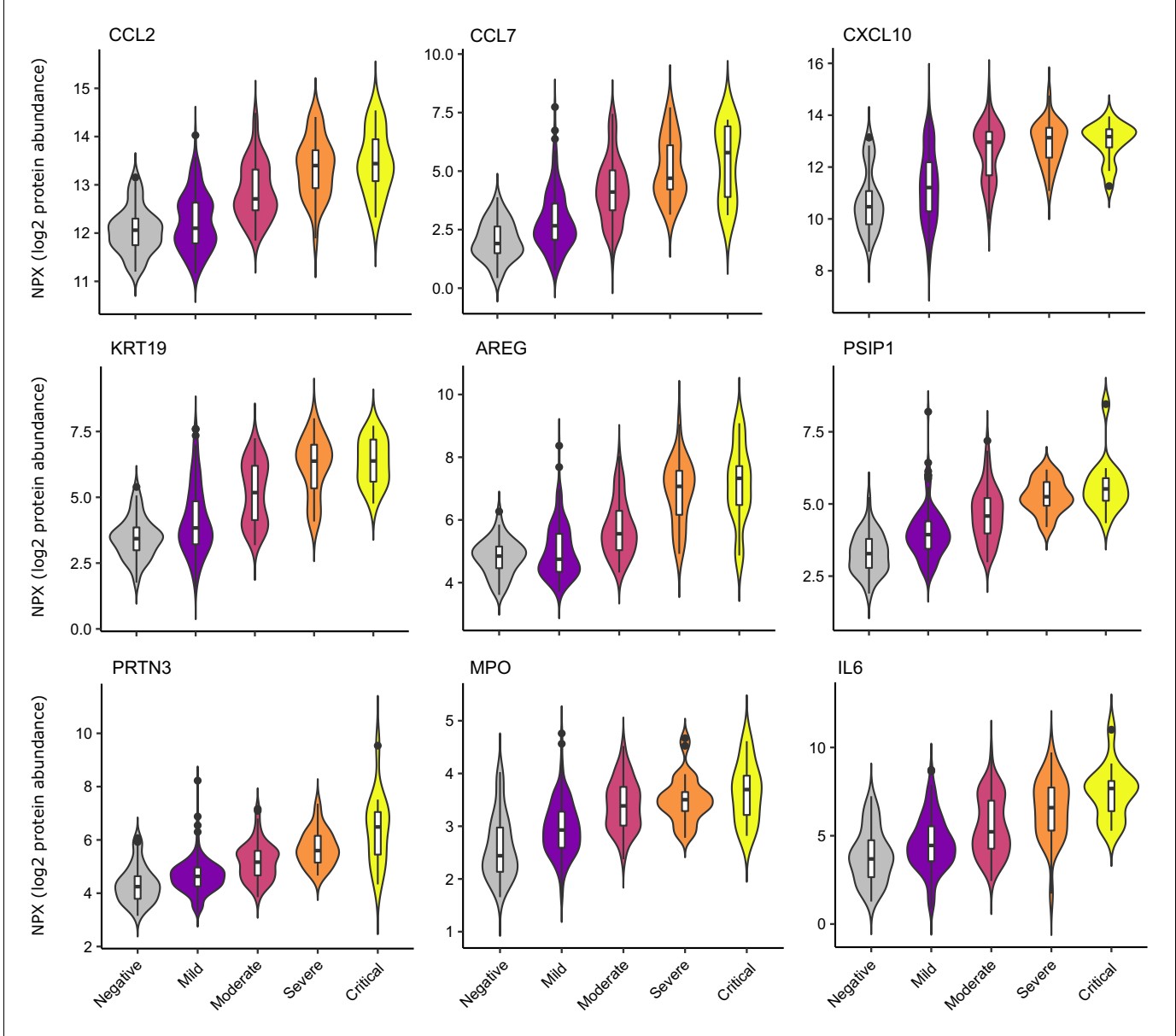

**Figure 5.** Selected proteins strongly associated with COVID-19 severity. Violin plots showing distribution of plasma protein levels according to COVID-19 status at the time of blood draw. Boxplots indicate median and inter-quartile range. n = 256 samples from 55 COVID-19 patients and 51 samples from non-infected patients. WHO severity indicates the clinical severity score of the patient at the time the sample was taken. Mild n = 135 samples; moderate n = 77 samples; severe n = 29 samples; critical n = 15 samples. Upper: monocyte chemokines. Middle: markers of epithelial injury. Lower: two neutrophil proteases and IL6.

calculating feature importance metrics (see Materials and methods, *Supplementary file 1e*). The most important proteins for indicating the presence of current or future severe disease were IL18BP, CTSD (Cathepsin D), GDF15, KRT19, TNFSF11, and IL1RL1 (ST2) (*Figure 7a*). It is notable that through this distinct analytical approach, KRT19 again emerged as a key biomarker of severe disease.

## Proteins associated with risk of death

Nine of 55 patients in subcohort A died. We therefore sought to identify proteins associated with risk of death. To leverage the dynamic nature of repeated protein measurements for prediction of death, we utilised joint models, which combine linear mixed models and Cox proportional hazards

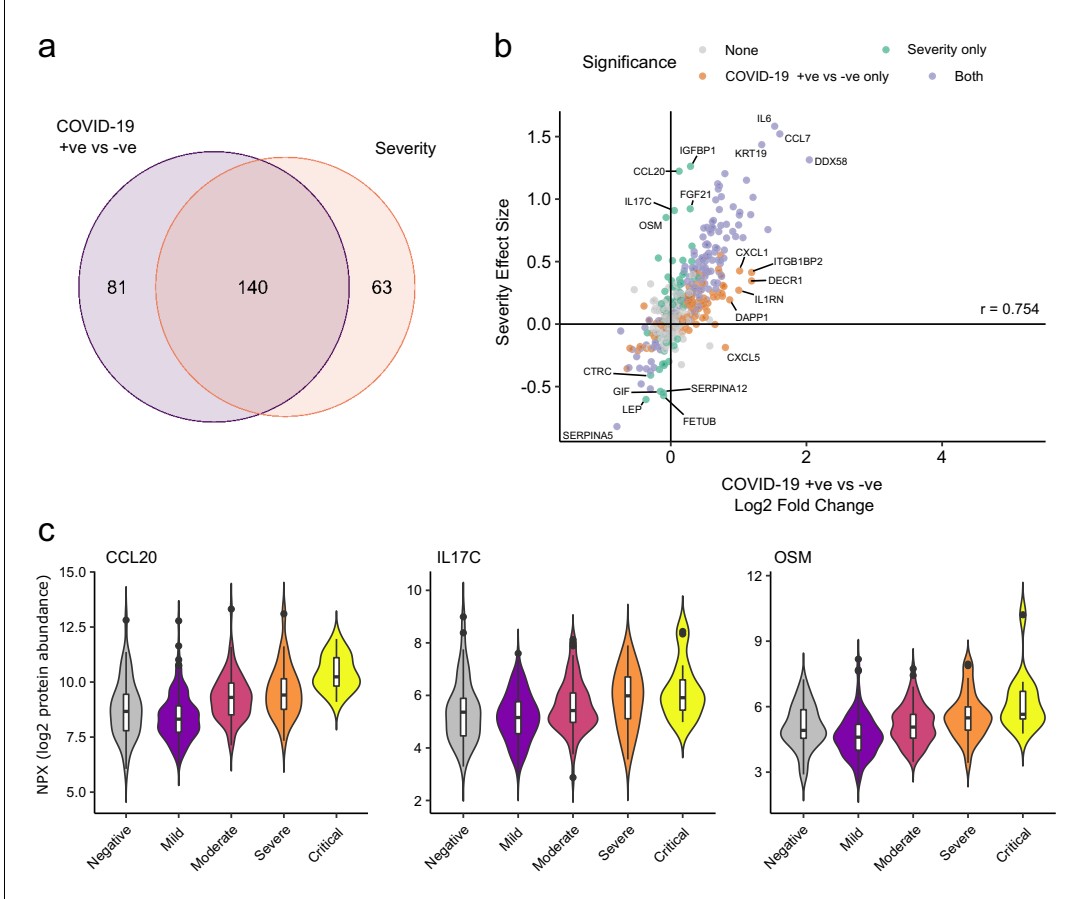

**Figure 6.** Comparison of proteins differentially expressed in COVID-19 with those associated with clinical severity. (a) Overlap between the proteins significantly differentially expressed in COVID-19 (n = 256 COVID-19 samples and 51 non-infected samples) versus those associated with severity (within-case analysis, n = 256 samples) (subcohort A). 5% FDR was used as the significant cut-off in both analyses. (b) Comparison of effect sizes for each protein in the COVID-19-positive versus -negative analysis (x-axis) and severity analysis (y-axis). Each point represents a protein. Pearson's r is shown. (c) Examples of proteins specifically associated with severity, but not significantly differentially abundant in the comparison of all cases versus controls. Violin plots showing distribution of plasma protein levels according to COVID-19 status at the time of blood draw. Boxplots indicate median and inter-quartile range. n = 256 samples from 55 COVID-19 patients and 51 samples from non-infected patients. WHO severity indicates the clinical severity score of the patient at the time the sample was taken. Mild n = 135 samples; moderate n = 77 samples; severe n = 29 samples; critical n = 15 samples.

models (*Ibrahim et al., 2010*; *Rizopoulos, 2010*) (see Materials and methods). This analysis identified 44 proteins for which increased concentration was associated with increased risk of death (*Figure 7b*, *Supplementary file 1f*), including CST3, IL22RA1, AZU1, CCL28, and SPON1, and 25 proteins for which increased concentration was associated with reduced risk of death, including CD84, TNFSF12, TANK, PRKCQ, and ADM.

## Associations with clinical laboratory tests

A number of routine clinical laboratory tests have well-characterised associations with COVID-19 (e.g. elevated inflammatory markers, d-dimer, and reduced lymphocyte count) (*Guan et al., 2020*). We therefore compared our proteomic data from COVID-19 patients at each timepoint to contemporaneous clinical laboratory measurements using linear mixed models (see Materials and methods). We found associations between plasma proteins and all clinical laboratory measurements except troponin (*Figure 8*, *Supplementary file 1g*). Many of these proteins were also markers of severity (e.g. IL6, KRT19, IFN-gamma, and CXCL10 were strongly associated with raised CRP and ferritin and reduced lymphocyte counts). Of note, CCL7, a monocyte chemokine that was also identified as an important marker of severity by the Random Forests classifier, was associated with lower monocyte

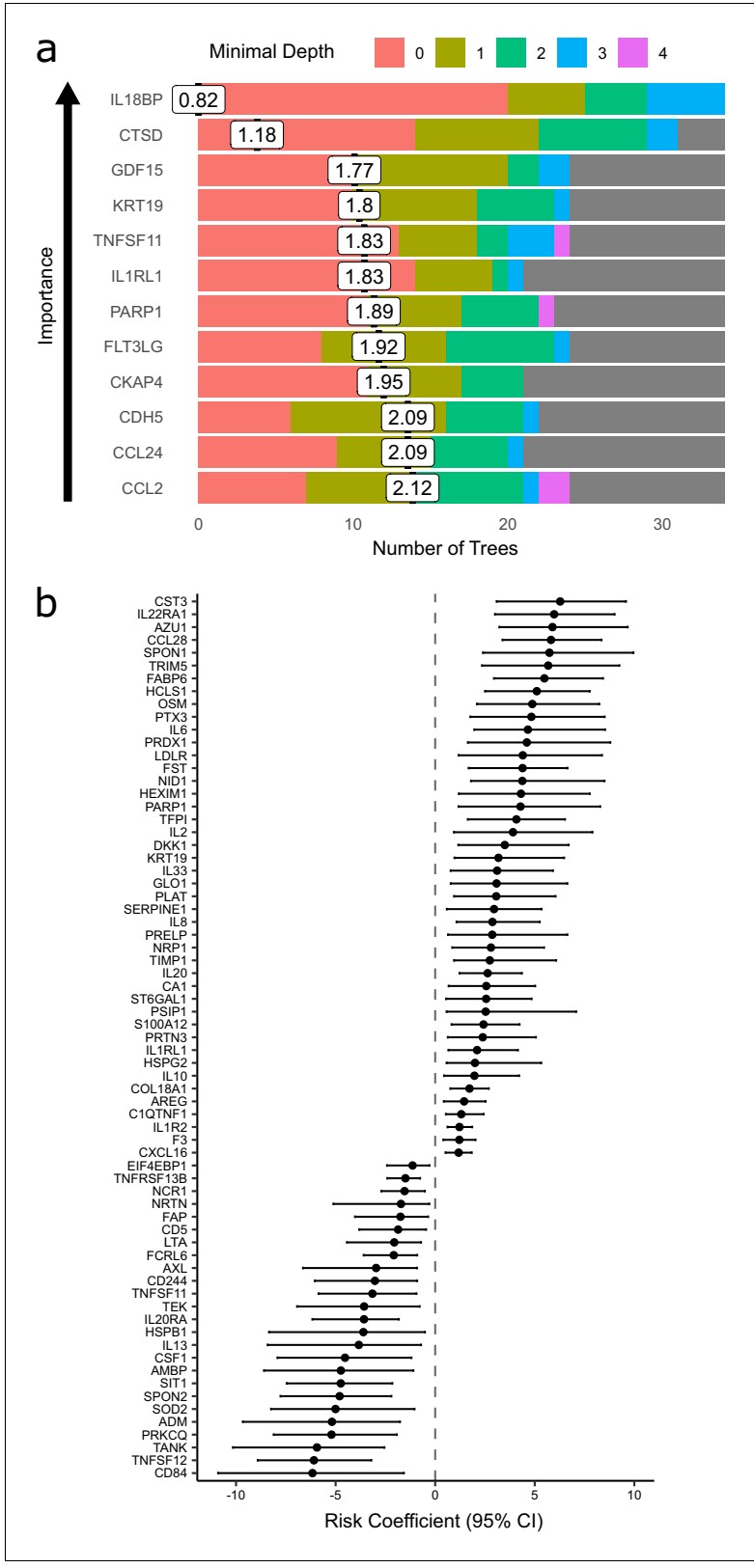

**Figure 7.** Prediction of severe COVID-19 and death. (a) The 12 most important proteins for predicting overall clinical course (defined by peak COVID-19 WHO severity) using Random Forests supervised learning. If a variable is important for prediction, it is likely to appear in many decision trees (number of trees) and be close to the root node (i.e. have a low minimal depth). The mean minimal depth across all trees (white box) was used as the primary

*Figure 7 continued on next page*

*Figure 7 continued*

feature selection metric. (**b**) Proteins that are significant predictors of death (Benjamini–Hochberg adjusted p<0.05). n = 256 samples from 55 COVID-19-positive patients, of whom nine died. Risk coefficient estimates are from a joint model. Bars indicate 95% confidence intervals. For proteins with a positive risk coefficient, a higher concentration corresponds to a high risk of death, and vice versa for proteins with negative coefficients.
The online version of this article includes the following figure supplement(s) for figure 7:

**Figure supplement 1.** Proteins associated with risk of death: correlation to clinical severity and clinical laboratory measurements.

count and raised inflammatory markers. Elevated neutrophil count was associated with Oncostatin-M, which regulates IL6, GCSF, and GMCSF production, and with the proteases MMP9 and defensin.

## Longitudinal analysis reveals proteins with distinct temporal profiles according to severity

The immune response to infection is dynamic, and therefore snapshot measurements provide only partial insights. Leveraging the dense serial sampling in our dataset (*Figure 1*), we modelled the temporal trajectory of each protein and asked whether or not any protein trajectories differed in patients with a severe/critical versus mild/moderate overall clinical course. This was achieved using linear mixed models that included a term for time from first symptoms and a time × severity interaction term (see Materials and methods).

One hundred and seventy-eight proteins displayed a significant association with time from first symptoms (5% FDR), demonstrating the temporal variability in plasma proteins across the disease course (*Supplementary file 1h*). Moreover, we identified 32 proteins for which there was significant interaction between time and severity, that is, proteins displaying differential temporal trajectories between mild/moderate and severe/critical infections (*Supplementary file 1h*, *Figure 9*). Among the proteins with the strongest temporal differences according to clinical course were the integrins ITGA11 and ITGB6, the adhesion molecule ICAM1, TNFRSF10B (a receptor for TRAIL), and PLAUR (the receptor for urokinase plasminogen activator). Most of these proteins exhibited rising profiles in the more severe patients but flat profiles in milder cases. ACE2, the receptor for SARS-CoV-2, also displayed this pattern (*Figure 9*). In contrast, abundance of ITGA11, which was also identified as reduced in the analysis of infected versus non-infected patients, fell over time in the severe group.

## Testing for proteins associated with ethnicity

In the UK, individuals from ethnic minorities are at higher risk of severe disease and death from COVID-19 (*Williamson et al., 2020*). We therefore examined whether any of the proteins we measured exhibited differences across ethnicities, analysing COVID-19-positive cases and controls separately (see Materials and Methods). In COVID-19-negative ESKD patients, no proteins were significantly associated with ethnicity in a multivariable model adjusting for age and sex. In COVID-19-positive ESKD patients, there is the potential for protein associations with ethnicity to be confounded by disease severity. To account for this, we included severity as well as age and sex as covariates. A single protein, LY75, was associated with ethnicity in this multivariable model (nominal p-value 0.0001, Benjamini–Hochberg adjusted p-value 0.04, with higher levels in white patients). Using the same within-case analysis strategy in subcohort B, we found no proteins were significantly associated with ethnicity after multiple testing correction, although the nominal p-value for LY75 was 0.025. While these analyses failed to identify substantial ethnicity-related variation in the proteins we measured, an important caveat is that there were relatively modest numbers of individuals from each ethnic group, and so statistical power was limited. Larger multi-ethnic studies are needed to adequately address this question.

## Comparisons to other proteomic studies in COVID-19

Other studies have used a variety of proteomic platforms to investigate COVID-19. We compared our findings to those of three published studies (*Shen et al., 2020*; *Lucas et al., 2020*; *Arunachalam et al., 2020*) and a preprint by *Filbin et al., 2020*. Of the 221 proteins that were

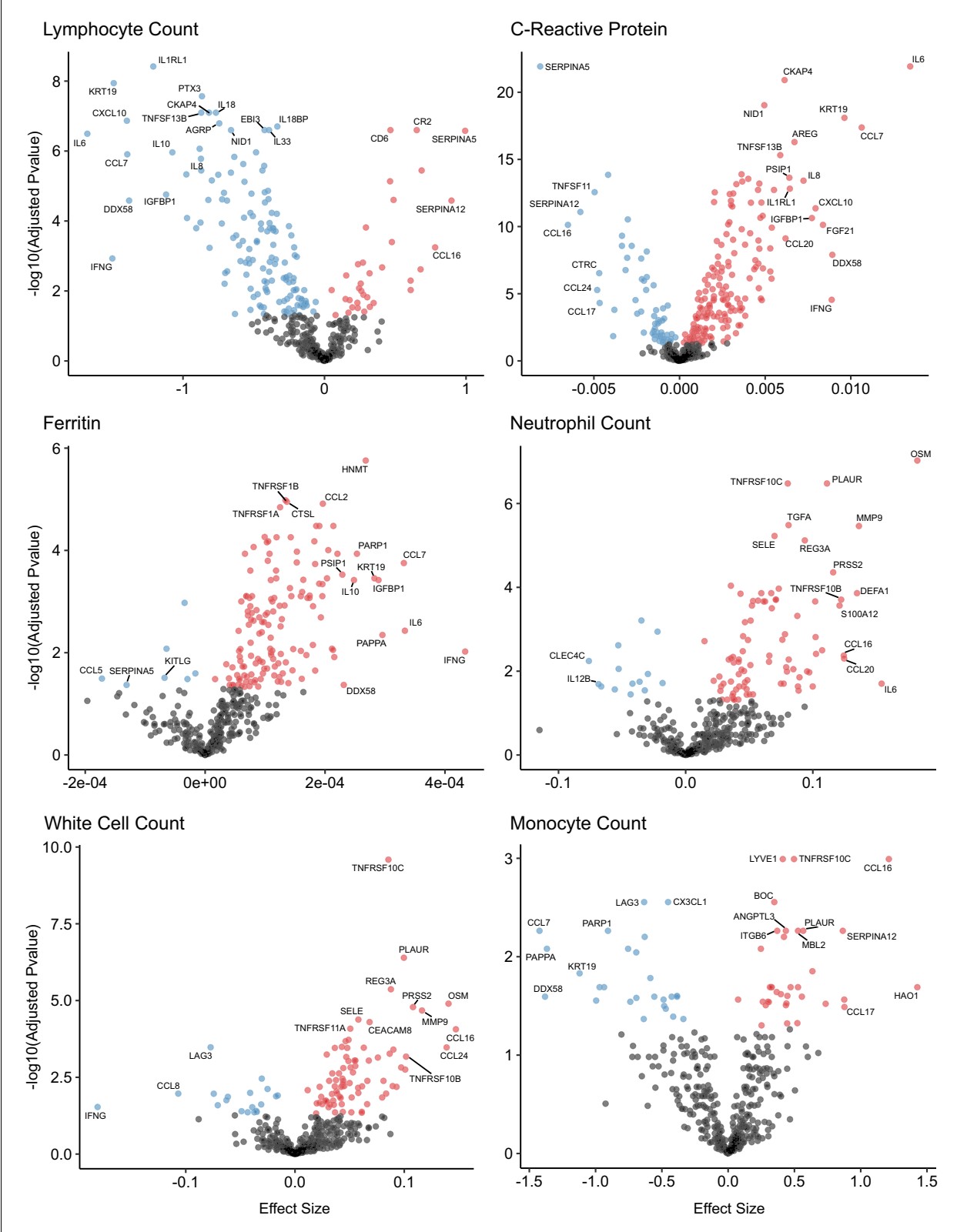

**Figure 8.** Associations of clinical laboratory markers with plasma proteins. Proteins that are positively (red) or negatively (blue) associated with clinical laboratory parameters (5% FDR). p-values from differential abundance analysis using linear mixed models after Benjamini–Hochberg adjustment. Dark-grey = non-significant. Two associations were found for d-dimer (not shown – see *Supplementary file 1g*).

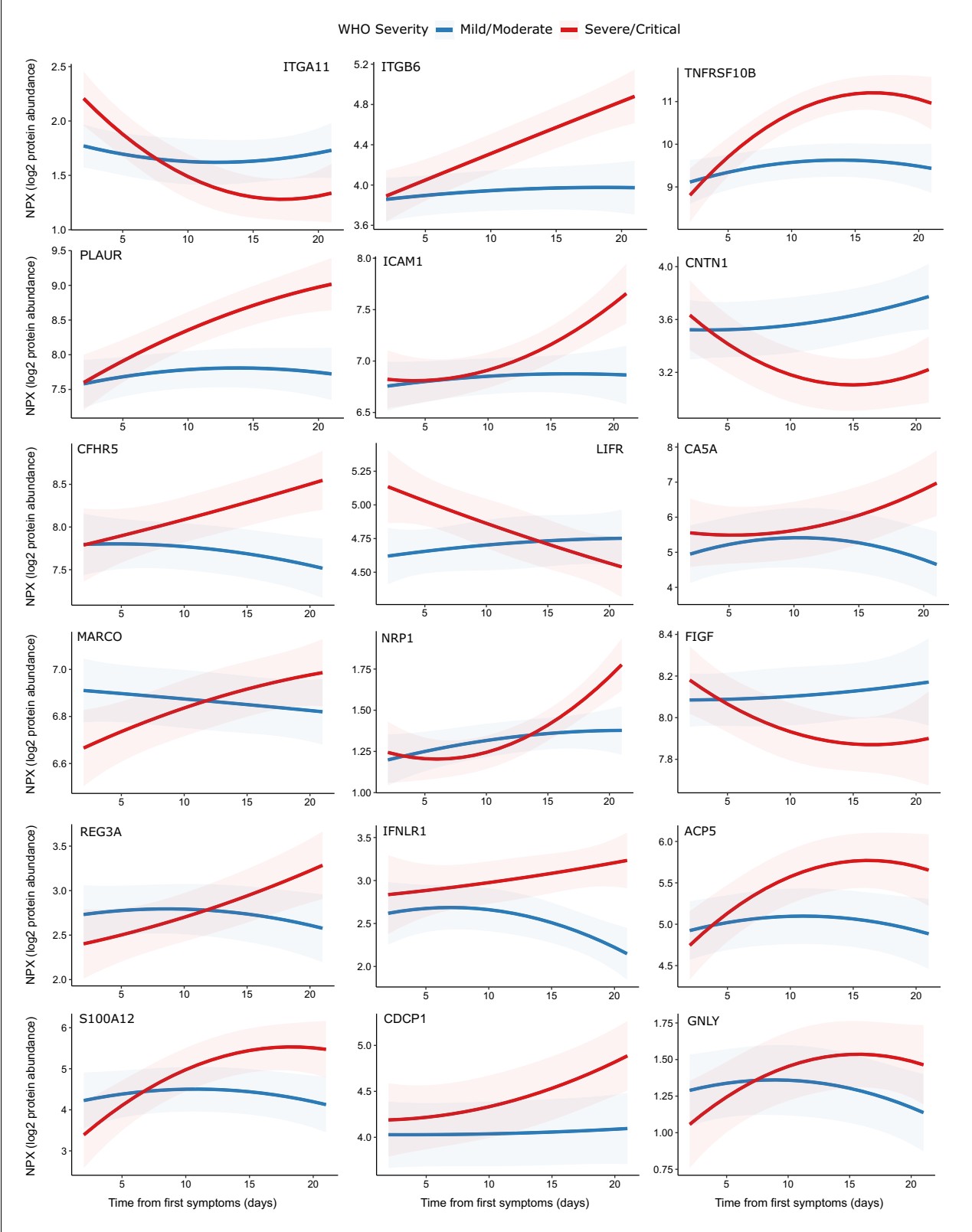

**Figure 9.** Modelling of temporal protein trajectories. The top 18 proteins displaying the most significantly (5% FDR) different longitudinal trajectories between patients with a mild or moderate (n = 28) versus severe or critical (n = 27) overall clinical course (defined by peak WHO severity). Means and 95% confidence intervals for each group, predicted using linear mixed models (see Materials and methods), are plotted. The remainder of significant proteins are shown in *Figure 9—figure supplement 1*. Individual data points are shown in *Figure 9—figure supplement 2*.

*Figure 9 continued on next page*

*Figure 9 continued*

The online version of this article includes the following figure supplement(s) for figure 9:

**Figure supplement 1.** Display of modelled temporal trajectories for other proteins with a significant time × severity interaction.

**Figure supplement 2.** Raw data points for modelling of temporal protein trajectories.

differentially abundant in our analysis of COVID-19-positive versus -negative ESKD patients, 116 associations had been previously reported (*Supplementary file 1i*). Of the 203 proteins associated with severity, 165 had previously been reported (*Supplementary file 1j*).

We focussed in more detail on the study by *Filbin et al., 2020* because of the large sample size and the breadth of proteomic assay used. This study comprised 384 patients with acute respiratory distress (306 COVID-19 positive and 79 COVID-19 negative) and measured 1472 proteins using the Olink Explore platform. Four hundred and seventeen of these were also measured in our study. Of the 221 proteins differentially abundant in our case/control analysis, 210 were measured in their study. Of these, 100 (47.6%) were significant in their analysis of COVID-19-positive versus COVID-19-negative respiratory distress. In addition, we observed strong correlation (r = 0.69) between the estimated log fold changes in our and their studies (*Figure 4—figure supplement 1*). Of the 203 proteins associated with severity in our study, 192 were measured in their study. One hundred and fifty-seven of these were significantly associated with severity, giving a concordance of 81.8%. Thus, despite the differences in study design and clinical populations, we observed notable similarities in our results and those reported by *Filbin et al., 2020*.

## Discussion

In this study, we performed plasma proteomic profiling of haemodialysis patients with COVID-19. A strength of our study was that we were able to perform serial blood sampling in both the outpatient and inpatient settings, including longitudinal samples from the same individual before and after hospitalisation. This was possible because haemodialysis patients are unable to fully isolate as they must continue to attend for regular dialysis sessions. Moreover, haemodialysis patients represent an important group since ESKD is one of the strongest risk factors for death from COVID-19 (*Williamson et al., 2020*; *Corbett et al., 2020*; *Ng et al., 2020*; *Valeri et al., 2020*). Data from the UK Renal Registry shows that 7 and 14 day mortality for COVID-19-infected in-centre haemodialysis patients was 11% and 19%, respectively (*COVID-19 Data, 2020*). Data from the Scottish Renal Registry estimates 30 day mortality following a positive COVID-19 test as 22%, and as of 31 May 2020, 28.2% of renal replacement therapy patients who had a positive COVID-19 test had died (*Scottish Renal Registry COVID-19 report, 2020*). In our local population of 1352 in-centre haemodialysis patients, 315 patients had tested positive for COVID-19 by the end of our study period (31 May 2020), of whom 53% required hospitalisation and 85 (27%) died. The OpenSAFELY study (*Williamson et al., 2020*) examined ~17 million UK primary care records and linked these to the UK COVID-19 mortality register. Patients with estimated glomerular filtration rate (eGFR) < 30 ml/min/1.73 m$^2$ had a hazard ratio (HR) for death of 3.56 after adjustment for age and sex.

In part, the high mortality from COVID-19 in ESKD patients likely reflects the fact that these patients are enriched for cardiometabolic traits that predispose to severe COVID-19. However, in multivariable analyses adjusting for these factors, impaired renal function remains an independent risk factor for severe COVID-19 (*Williamson et al., 2020*). Moreover, there is an inverse relationship between renal function and risk of death from COVID-19 across the spectrum of chronic kidney disease. These observations support the notion that the state of ESKD per se is an important determinant of outcome in COVID-19. ESKD is well-recognised as an immunosuppressed state (*Eiselt et al., 2016*; *Girndt et al., 1999*; *Sarnak and Jaber, 2000*), with defects in both innate and adaptive immunity (*Alexiewicz et al., 1991*; *Massry and Smogorzewski, 2001*; *Girndt et al., 2001*; *Meier et al., 2002*). Accordingly, ESKD confers increased vulnerability to viral infections including influenza and respiratory syncytial virus (*Betjes, 2013*; *Boattini et al., 2020*; *Prasad et al., 2020*; *Usvyat et al., 2012*). In addition, ESKD results in a chronic low-grade inflammatory state (*Carrero and Stenvinkel, 2010*). This tendency to a pro-inflammatory state, combined with reduced ability to respond to

viruses, may contribute to the abnormal host response to SARS-CoV-2 infection, producing the immunopathology that leads to severe COVID-19.

Our comparison of COVID-19-positive and -negative haemodialysis patient plasma samples revealed 221 proteins that were differentially abundant in COVID-19. The majority of these were upregulated, with strong representation of viral response proteins (e.g. DDX58, IFNG), cytokines/ chemokines (e.g. IL6, CCL7, CXCL10, and CXCL11), and epithelial proteins (e.g. KRT19, PSIP1) (*Figure 3a*). The COVID-19-negative controls in this analysis were carefully matched to cases in terms of age, sex, and ethnicity. However, complete matching of clinical characteristics was not feasible; there were differences in the prevalence of diabetes and the underlying causes of ESKD between COVID-19-positive cases and controls (*Table 1*). Sensitivity analyses adjusting for these covariates gave highly consistent results, indicating that our findings are robust. In addition, we validated our findings when we analysed serum samples from a separate subcohort of COVID-19-positive ESKD patients.

ESKD is itself likely to significantly impact the plasma proteome. Previous cross-sectional studies have shown that the levels of many circulating proteins have an inverse relationship with eGFR (*Naseeb et al., 2015*; *Christensson et al., 2018*). A longitudinal study using an Olink proteomics panel (although not one used in our study) found that for 74% of the 84 proteins measured, protein levels rose as eGFR fell (*Lind et al., 2019*). For many proteins, it is unclear whether this inverse relationship with renal function reflects cause or effect. Some proteins may be increased in chronic kidney disease due to reduced renal clearance, some may be elevated secondary to tissue injury or chronic inflammation, and others may be drivers of renal injury. Regardless, this observation of widespread changes in the blood proteome of kidney disease patients emphasises the importance of using COVID-19 ESKD patients rather than healthy individuals as our control group.

Analysis within COVID-19 cases revealed 203 proteins associated with disease severity, the strongest of which was IL6 (*Figure 3b*). Association of IL6 with severe disease is well-established and has already received considerable attention (*Wu et al., 2020*; *Li et al., 2020*). Despite promising initial case reports of IL6R receptor blockade in COVID-19, convincing efficacy was not demonstrated in early randomised trials (*Furlow, 2020*). More recently, the REMAP-CAP trial has shown the benefit of anti-IL6R therapy when given to critically ill patients on admission to intensive care units (*Gordon et al., 2021*), indicating that IL6 does contribute to critical illness from COVID-19. Our finding that IL6 was most strongly upregulated in severe disease demonstrates the value of plasma proteomic profiling in identifying putative drug targets.

Members of the CCL and CXCL chemokine families (e.g. CCL2, CCL7, CCL20, and CXCL10) were strongly associated with severity. Likewise, higher levels of CCL2, CCL7, CCL20, and CXCL10 were associated with lower blood lymphocyte count and higher inflammatory markers (CRP and ferritin) (*Supplementary file 1g*), which are clinical markers of severe disease and poorer outcome in COVID-19 (*Gupta et al., 2021*). Of note, CCL20 is a chemoattractant for lymphocytes (*Schutyser et al., 2003*), and its negative association with lymphocyte count may reflect a direct effect on migration of lymphocytes from the blood into the tissues rather than simply marking severe disease. CCL2 (also known as MCP-1) and CCL7 (MCP-3) are both chemokines for monocytes, and CXCL10 has pleiotropic immunological effects including chemotaxis. These chemokines were also negatively correlated with blood monocyte count, suggesting recruitment of these innate immune cells into damaged tissues.

The neutrophil proteases PRTN3 (proteinase-3) and MPO (myeloperoxidase) (*Figure 5*) and the neutrophil-derived protein AZU1 were associated with severe disease (*Supplementary file 1d*), indicating that neutrophil activation and degranulation are features of severe COVID-19. Degranulation of neutrophils releasing PRTN3 and MPO could potentially contribute to oxidative damage in the lungs and thus more severe disease.

A striking finding of our study was the association of disease severity with upregulation of epithelial proteins (e.g. KRT19) and epithelial tissue repair pathways (e.g. PSIP1, AREG, GRN [progranulin]), most likely reflecting lung and vascular damage. KRT19 was notably prominent in our analyses, as well as the study by *Filbin et al., 2020*; *Supplementary file 1j*. KRT19 is an intermediate filament protein, important for the structural integrity of epithelial cells (*Saha et al., 2017*). These data suggest that severe COVID-19 is characterised by destruction of the lung epithelium and vascular endothelium. Vascular injury might thus explain the high level of vascular thrombosis seen in patients in

severe disease. In summary, our data reveal that severe COVID-19 is characterised proteomically by a signature of innate immune activation and epithelial injury.

Sixty-nine percent of proteins associated with severity were also differentially abundant in the case versus control analysis (*Figure 6a*), and for the large majority of proteins the within-case severity analysis, effect size was proportional to the fold change between cases and controls (*Figure 6b*). This suggests that, in general, the distinction in the plasma proteome between severe and mild COVID-19 is a quantitative difference in the COVID-19 signature, rather than there being an orthogonal signature involving a different set of proteins. Consistent with this concept, examination of PCA plots coloured by severity revealed that while there was a gradient of COVID-19 severity, the samples from severe or critical patients did not form a discrete cluster distinct from those from patients with milder disease (*Figure 2—figure supplement 1*). However, there were a few exceptions where proteins that were associated with severity were not upregulated in the case–control analysis. These included OSM, IL17C, and CCL20 (*Figure 6c*). These proteins therefore reflect biological processes specifically of severe disease and may represent therapeutic targets.

Survival analysis identified 44 proteins associated with increased risk of death (*Figure 7b*). As expected, many of these were also associated with disease severity, high CRP, and lower lymphocyte count (*Figure 7—figure supplement 1*). In contrast, 25 proteins were associated with reduced risk of death (*Figure 7b*). One such protein is the multi-functional cytokine TNFSF12 (TWEAK). Although TWEAK can exert pro-inflammatory effects, it also can inhibit the innate immune response (*Maecker et al., 2005*) and promote tissue repair and endothelial cell proliferation and survival (*Burkly et al., 2007*), which may be beneficial responses in COVID-19. This illustrates that although proteins associated with inflammation are often thought to be destructive, the inflammatory response also induces programmes for limiting injury and initiating tissue repair. Insufficient activation of such homeostatic mechanisms may contribute to why some individuals get severe COVID-19.

The host immune response to COVID-19 is a dynamic process, and clinical deterioration typically occurs 7–10 days after first symptoms. Temporal information may therefore be important in determining optimum timing of therapeutic intervention (e.g. blockade of a particular cytokine). By taking serial samples and examining their patterns within individuals over time, we were able to model protein trajectories and found that many proteins display temporal variability during COVID-19. Longitudinal measurements also allow molecular comparison of severe versus mild disease trajectories. By modelling the interaction term between time from first symptoms and overall disease course, we found 32 proteins that displayed distinct temporal profiles in severe versus mild disease. These results point to enhanced leucocyte–endothelial cell interactions indicated by upregulation of cell adhesion molecules (e.g. ITGB6, ICAM1) in severe disease. This endothelial activation may contribute to COVID-19-associated thrombosis discussed above. Management of thrombosis in COVID-19 currently consists of anticoagulation. Our results suggest that disrupting leucocyte–endothelial interactions may be a complementary therapeutic strategy.

Several proteins associated with either risk of death or clinical severity lie in pathways targeted by existing drugs. PARP1 was identified as an important marker of current or future severe COVID-19 and also was associated with risk of death. PARP1 is associated with inflammatory and vascular disease (*Henning et al., 2018*). PARP1 inhibitors are in use for cancer (*Rouleau et al., 2010*), and our data suggest that re-purposing of PARP1 inhibition in COVID-19 should be explored further. IL33 was associated with both risk of death and clinical severity, and its receptor IL1RL1 (ST2) was associated with clinical severity and identified as an important predictor of severe clinical course. Monoclonal antibodies against IL33 and its receptor are in late-stage development for asthma (*Corren, 2019*) and could also be explored in COVID-19. As discussed above, MPO was associated with clinical severity. MPO inhibitors (*Galijasevic, 2019*) might have a role in reducing neutrophil-mediated tissue injury in COVID-19. Finally, inhibitors of monocyte chemokines (e.g. CCL2) and their receptors have been developed (*Vergunst et al., 2008*; *Haringman et al., 2006*), although drugging these pathways is made more challenging by molecular cross-talk. An important caveat is that we cannot determine whether the associations we observed are drivers of pathology in COVID-19 or simply reflect the downstream consequences of inflammation and tissue injury. Future studies using Mendelian randomisation analysis will provide a useful tool for assessing causality and prioritising drug targets.

Other groups have studied the plasma or serum proteome in COVID-19 (*Shen et al., 2020*; *Lucas et al., 2020*; *Arunachalam et al., 2020*; *Filbin et al., 2020*; *Rodriguez et al., 2020*), using

either mass spectrometry or immunoassays including the Olink platform. Mass spectrometry is less sensitive than immunoassays and so it is likely to be unable to detect many of the cytokines measured here. Conversely, it can provide complementary information by measuring many proteins that our immunoassays did not target. A limitation of our study was that we used Olink panels that measured specific proteins selected on their relevance to inflammation, immunity, cardiovascular, and metabolic disease. This bias precluded formal pathway enrichment analysis of differentially abundant proteins. In general, our results had greater similarities to studies that used immunoassays over mass spectrometry (*Supplementary file 1i,j*). 47.6% of proteins differentially expressed in COVID-19-positive versus -negative ESKD patients in our study were differentially expressed in COVID-19-positive versus -negative acute respiratory distress syndrome patients in the study of *Filbin et al., 2020*, who used a different Olink proteomics platform. Moreover, we observed consistent effect sizes (*Figure 4—figure supplement 1*). These similarities are striking given the difference in clinical populations and control groups; in Filbin et al.'s report, the controls included patients with non-COVID-19 respiratory infections, whereas our control group did not have active infection. The concordance in proteins associated with COVID-19 severity within cases was even higher (81.8%). The similarities suggest a similar plasma proteomic signature of COVID-19 across different clinical populations, particularly the signature associated with severity.

In summary, this study reveals proteins associated with COVID-19 infection and severity and demonstrates altered dynamic profiles between patients with severe disease and those with a more indolent course. Our results emphasise the importance of studying and targeting mechanisms that reduce the lung epithelial and endothelial damage to both alleviate the severity of the infection and reduce the chance of long-lasting complications. These data provide a valuable resource for therapeutic target prioritisation.

## Materials and methods

### Subjects and samples

#### Ethical approval

All participants (patients and controls) were recruited from the Imperial College Renal and Transplant Centre and its satellite dialysis units, London, United Kingdom, and provided written informed consent prior to participation. Study ethics were reviewed by the UK National Health Service (NHS) Health Research Authority (HRA) and Health and Care Research Wales (HCRW) Research Ethics Committee (reference 20/WA/0123: The impact of COVID-19 on patients with renal disease and immunosuppressed patients). Ethical approval was given.

#### Subcohort A

We recruited 55 COVID-19-positive haemodialysis patients, either as outpatients or as inpatients (*Table 1*). All patients were receiving in-centre outpatient haemodialysis prior to COVID-19 diagnosis. COVID-19 was confirmed in all cases with positive nasal PCR for the SARS-CoV-2 virus. Patients were recruited during the first UK national lockdown, with recruitment from 8 April to 30 May 2020. Blood was collected in EDTA tubes and centrifuged to obtain plasma and stored at –80°C. Sample processing was performed within 4 hours of venepuncture. The initial sample was taken as an outpatient for 30 patients and as an inpatient for 25 patients. Where feasible, serial blood samples were taken. In total, 259 samples were taken (three subsequently failed QC – see below). The median number of serial samples was 5 (range 1–10) (*Figure 1—figure supplement 1d*). Eight patients who were recruited as outpatients were subsequently admitted to hospital with COVID-19 over the course of the study. Twenty-seven of 55 (49.1%) patients had severe or critical disease (defined by peak WHO severity). Nine (16.4%) patients died.

In addition, we recruited 51 COVID-19-negative haemodialysis controls. COVID-19-negative haemodialysis controls were selected to mirror the cases in terms of demographic features (age, sex, ethnicity) (*Figure 1—figure supplement 1a–c*). These control patients had no clinical features of any other infection.

## Subcohort B

We also recruited a separate set of 46 COVID-19-positive haemodialysis patients (*Table 2*). These patients were recruited from the same centre, but slightly earlier than subcohort A (recruitment commenced on 30 March 2020). For these patients, blood was collected in serum tubes and centrifuged to obtain serum. At this time, we had very limited access to laboratory facilities, and so plasma was not collected from these patients. Five were outpatients and 41 were inpatients, reflecting the fact that UK policy was weighted towards inpatient testing at the time these patients were recruited. Thirty-three of 46 patients (71.7%) had severe or critical disease (by peak WHO severity), and nine (19.6%) patients died. For 40 patients, only one sample from a single timepoint was collected, and for six patients, two samples were collected. To provide controls for subcohort B, we used serum samples from 11 non-infected haemodialysis patients (collected at the same time as plasma from a subset of the ESKD control group described above).

## Clinical severity scores

Severity scoring was performed based on WHO classifications (WHO clinical management of COVID-19: Interim guidance 27 May 2020) adapted for clinical data available from electronic medical records. 'Mild' was defined as COVID-19 symptoms but no evidence of pneumonia and no hypoxia. 'Moderate' was defined as symptoms of pneumonia or hypoxia with oxygen saturation ($SaO_2$) greater than 92% on air, or an oxygen requirement no greater than 4 L/min. 'Severe' was defined as $SaO_2$ less than 92% on air, or respiratory rate more than 30 per minute, or oxygen requirement more than 4 L/min. 'Critical' was defined as organ dysfunction or shock or need for high dependency or intensive care support (i.e. the need for non-invasive ventilation or intubation). Severity scores were charted throughout a patient's illness. We defined the overall severity/clinical course for each patient as the peak severity score that occurred during the patient's illness.

## Proteomic assays

Plasma and serum proteomic measurements were performed using Olink proximity extension immunoassays (https://www.olink.com/products/). Five 92-protein multiplex Olink panels were run ('inflammation', 'immune response', 'cardiometabolic', 'cardiovascular 2', and 'cardiovascular 3'), resulting in 460 measurements per sample. Since a small number of proteins were measured on more than one panel, we measured a total of 436 unique proteins. The Olink assays were run using 88 samples/plate. All plates were run in a single batch. Plate layouts was carefully designed to avoid confounding of potential plate effects with biological or clinical variables of interest. To achieve this, we used an experimental design that combined ensuring case/control balance across plates with random selection of samples from each category and random ordering of allocation to wells. This is outlined in more detail as follows. We ensured that each plate contained a mixture of control and case samples. Specifically, a fixed proportion of each plate was designated for control samples. The allocation of specific control samples to each plate was performed using randomisation. For the case samples, we again used randomisation for plate assignment, with the constraint that once one sample from a given patient was allocated to a plate, all other longitudinal samples from that patient were assigned to same plate. Finally, once all the samples had been allocated to plates, the layout of samples within each plate was determined through a further randomisation step for well allocation.

## Protein annotation

We used the Human Protein Atlas version 20.0 (*Uhlén et al., 2015*) for protein annotation (*Figure 1—figure supplement 1*). We performed enrichment analysis of the 436 proteins that we measured using string-db (*Szklarczyk et al., 2019*).

## Normalisation and quality assessment and control

The data was normalised using standard Olink workflows to produce relative protein abundance on a log2 scale ('NPX'). Quality assessment was performed by (1) examination of Olink internal controls and (2) inspection of boxplots, relative log expression plots (*Gandolfo and Speed, 2018*), and PCA. Following these steps, three poor-quality samples were removed. In addition, five samples failed QC on a single proteomic panel only, with the remaining panels passing QC. For these samples,

proteins on the panel that failed QC were set to missing, and the data for the remaining proteins was retained.

PCA revealed no substantial impact of plate effects (*Figure 2—figure supplement 2*). Thirteen proteins were assayed more than once due to their inclusion in multiple Olink panels. For plasma, the median correlation between the assays was 0.986 with an inter-quartile range (IQR) of 0.974–0.993 and a range of 0.925–0.998. For serum, the median correlation between the assays was 0.991 with an IQR of 0.952–0.995 and a range of 0.737–0.999. We removed duplicate assays at random prior to subsequent analyses.

For 11 ESKD controls, we had contemporaneous plasma and serum samples. To assess the comparability of these two matrices, we calculated the Pearson's correlation coefficient between the assays for each protein (*Supplementary file 1k*). Three hundred and forty-four of 436 (78.9%) proteins had a Pearson's r > 0.5. We also report the variance of each protein in plasma and serum since low correlation may reflect low variance. The proteins with the lowest estimated Pearson correlation coefficient were AZU1, STK4, and TANK. We highlight that this comparison had small sample size (only 11 samples) and that the samples were from control patients without infection. Caution should be made in extrapolating these findings to the context of active infection where protein dynamic ranges may be different.

## Missing values

Following QC, 0.22% data points were missing for the plasma dataset and 0.35% for the serum dataset. For analyses that required no missing values (PCA and supervised learning), we imputed missing values as follows. The dataset was first scaled and centred, and missing values imputed using caret's k-nearest neighbours method (*Kuhn, 2008*). The five closest samples (by Euclidean distance) were used to estimate each missing value.

## Principal component analysis

Singular value decomposition was used to perform PCA on the proteomic data from subcohort A (plasma samples). We then used the loadings from subcohort A together with the proteomic data from subcohort B to calculate principal component scores. This enabled projection of subcohort B data into the PCA space of subcohort A.

## Differential protein abundance analysis: COVID-19 positive versus negative

Differential protein abundance analyses between COVID-19 positive and negative samples were performed using linear mixed models, to account for the use of serial samples from the same individuals (R lme4 package *Bates et al., 2015*). This analysis compared 256 samples from 55 COVID-19 patients with 51 non-infected patients (one sample per non-infected patient). Age, sex, and ethnicity were included as covariates. We used a random intercept term to estimate the variability between individuals in the study and account for repeated measures. The regression model in R notation was:

$$NPX \sim covid\_status + sex + age + ethnicity + (1|individual)$$

where NPX represents the protein abundance and covid_status was a categorical variable (infected/non-infected). Sex and ethnicity were also categorical variables. Age was a quantitative variable. We calculated P-values using a type 3 F-test in conjunction with Satterthwaite's method for estimating the degrees of freedom for fixed effects (*Kuznetsova et al., 2017*). The regression model was fitted for each of the 436 proteins individually. Multiple testing correction was performed using the Benjamini–Hochberg method and a 5% FDR used for the significance threshold.

The same approach was used for subcohort B. This analysis comprised 52 serum samples from 46 COVID-19-positive patients versus 11 samples from non-infected patient samples (one sample per non-infected patient).

As sensitivity analyses, we repeated the differential abundance analyses between case and controls for the subcohort A adjusting for additional covariates and comparing this to the basic model (i.e. using age, sex, and ethnicity alone). This was performed for each of the following parameters: diabetes status, cause of ESKD, and time to last haemodialysis.

## Testing for associations between proteins and clinical severity

For testing the association of plasma proteins with the four-level WHO severity rating (mild, moderate, severe, and critical) within COVID-19-positive cases from subcohort A (n = 256 samples from 55 patients), we used a similar linear mixed modelling approach to the COVID-19-positive versus -negative differential abundance analysis; for this analysis, the covid_status term was replaced by a severity variable encoded using orthogonal polynomial contrasts to account for ordinal nature of severity levels. As before, age, sex, and ethnicity were included as covariates. As a sensitivity analysis, we repeated the analysis with time to last haemodialysis as an additional covariate.

## Testing for associations between proteins and clinical laboratory tests

The linear mixed modelling strategy was also employed for testing association of temporal clinical laboratory variables and protein levels, with the value of the clinical variable (as a quantitative trait) used in place of covid_status. Only COVID-19-positive patients were included in this analysis. Contemporaneous lab measurements were not available for all samples. This varied according to the clinical lab parameter. Some (e.g. troponin, d-dimer) were measured less frequently than full blood count and CRP. Details of the proportion of missing values for each lab parameter are included in *Supplementary file 1g*. We also calculated correlations between clinical laboratory variables and protein levels using the R package rmcorr, which determines the overall within-individual relationship among paired measures that have been taken on two or more occasion (*Bakdash and Marusich, 2017*).

## Testing for associations between proteins and ethnicity

We performed testing of protein levels and ethnicity separately in COVID-19-negative ESKD patients and COVID-19-positive ESKD patients. These analyses were limited to individuals who were White, South Asian (Indian, Pakistani, or Bangladeshi ancestry), or Black as there were too few individuals from other ethnic groups for meaningful interpretation. For COVID-19-negative patients (one sample per patient), we performed linear regression for each protein with ethnicity as the predictor variable and age and sex as covariates. For COVID-19-positive patients, we used a linear mixed model to account for serial samples from the same individual, again with age, and sex as covariates.

## Multiple testing correction

We used the Benjamini–Hochberg method to control the FDR at 5% for all statistical analyses.

## Alternative estimation of the FDR using the plug-in method

To provide additional support that the Benjamini–Hochberg procedure was providing adequate control of the FDR, we also used the plug-in procedure (*Hastie et al., 2001*) as an alternative method to estimate the FDR, as described below.

1. We defined R as the number of associations declared significant in the real data.
2. We defined C as the test statistic used as the significance threshold used in the real data (i.e. that corresponding to an adjusted p-value of 0.05).
3. The expected number of proteins that we would find significant under the null hypothesis that no proteins are differentially abundant between COVID-19-positive versus -negative patients (i.e. false positives) was estimated using a permutation strategy. We randomly permuted each individual's COVID-19 status label 100,000 times and, in each case, repeated the differential abundance analysis on the permuted data. The estimated the number of false positives ($\hat{V}$) was then estimated by the number of associations with test statistic > C in 100,000 permutations of the data, divided by the number of permutations.
4. The estimated FDR was then calculated as $\hat{V}/R$.

We implemented a similar approach for the testing the association of proteins with severity scores within cases.

Using this method, the estimated FDR for the case versus control analysis was 0.062 and for the severity analysis 0.057, indicating that we had appropriately controlled the FDR.

## Empirical p-value calculation

As a complementary analysis, based on the approach of *Filbin et al., 2020*, we estimated the empirical p-value for the likelihood of observing as many significant proteins as we identified in the real data if the null hypothesis of no differentially abundant proteins in cases versus controls were true. We again used 100,000 permutations of the case–control labels to estimate the null distribution. We performed Benjamini–Hochberg adjustment on the nominal p-values of each permutation and counted the number of proteins that were significant (adjusted p-value<0.05) in each permutation.

The distribution of the number of proteins declared significant is shown in *Figure 3—figure supplement 2a*; on no occasion in 100,000 permutations did we observe more proteins declared significant than in the real data. We can thus state that the empirical p-value (the fraction of permutation runs where we observed $\geq$ the number of associations in the real data) is less than 1/100,000 = $1 \times 10^{-5}$.

We also applied this method to the association testing of proteins with severity scores within cases (*Figure 3—figure supplement 2b*). Again, on no occasion in 100,000 permutations did we observe more proteins declared significant than in the real data (empirical p-value<$1 \times 10^{-5}$).

## Supervised learning

Random forest models were fitted using R's randomForest and caret packages (*Kuhn, 2008*; *Leo, 2001*). Data was centred, scaled, and imputed as in *Missing values* with the caveat that, during cross-validation, the pre-processing procedure was first applied on the resampled (training) data before the same method was applied without re-calculation to the holdout (test) set. To estimate model accuracy, we used fourfold cross-validation. The cross-validation procedure was repeated 100 times. The model's parameters were kept constant at 500 trees and an mtry value (number of proteins randomly sampled as candidates at each node) calculated as the square root of the number of features. After parameter estimation, we fitted a final model trained using the entirety of the dataset. This model was used for subsequent feature extraction. Random forest feature extraction was carried out using the R randomForestExplainer package. We made use of the following importance measures: accuracy decrease (the average decrease in prediction accuracy upon swapping out a feature), number of trees (the number of trees with a node corresponding to a feature), and mean minimal depth (the average depth at which a node corresponding to a feature occurs). Three models were generated with different input features: (1) proteomic data alone; (2) clinical parameters alone; and (3) proteomic data and clinical parameters. Clinical parameters included sex, age, ethnicity, cause of ESKD, comorbidities, smoking status, radiological evidence of pulmonary infiltrates, and clinical laboratory tests.

## Survival analysis using joint modelling

Following scaling and centring, we fitted linear mixed models for each protein to capture the temporal trajectories of each individual. A polynomial spline of degree two was used to model protein concentration with respect to time (from symptom onset, measured in days); the spline was fitted for samples that were taken between 1 and 28 days from first symptoms, inclusive. Proteomic data after that point was censored. We estimated both random intercepts and random slopes for each individual, as per the following R formula notation:

$$\mathrm{NPX} \sim \mathrm{time} + (\mathrm{time}|\mathrm{individual})$$

These were joined to a Cox regression model using the jointModel package (*Rizopoulos, 2010*) in order to estimate the association of each protein with risk for death. P-values were calculated using a Wald test for the association between the linear mixed model and Cox regression. Benjamini–Hochberg adjustment was applied, with an adjusted p-value of 0.05 used as the significance threshold.

## Longitudinal analysis

We also used linear mixed models to estimate the temporal profile of each protein. For this longitudinal analysis, we explicitly modelled the time from first symptoms. We set up the model to test for each protein (1) whether the protein significantly change over time and (2) whether the protein changes over time differently in individuals with a mild versus severe disease course. The latter was

performed statistically by testing for an interaction effect between time and clinical course. For the purposes of this analysis, we binarised patients into severe or non-severe clinical course according to the peak WHO severity disease of their illness. Patients with a peak WHO score of mild or moderate were considered non-severe and those with a peak score of severe or critical were considered severe.

We then used R's bs function to fit a polynomial spline of degree two to model protein concentration with respect time (from symptom onset, measured in days) (*Perperoglou et al., 2019*). The spline was fitted for samples that were taken between 1 and 21 days from symptom onset, inclusive. We estimated random slopes with respect to time, in addition to random intercepts, to account for each individual's unique disease course. For each protein, we fitted the following model (R notation):

$$\text{NPX} \sim \text{time} * \text{severity} + \text{sex} + \text{age} + \text{ethnicity} + (\text{time}|\text{individual})$$

To identify proteins that changed significantly over time, we examined the p-values for the main effect of time. To identify proteins with distinct temporal profiles between severe and non-severe cases, we examined the p-values for the time × severity interaction term. For each of these two research questions, p-values were adjusted for the multiple proteins tested using the Benjamini–Hochberg method and 5% FDR used as the significance threshold.

## Code availability

Code is available in the following GitHub repository: https://github.com/jackgisby/longitudinal_olink_proteomics; *Gisby, 2021*; copy archived at swh:1:rev:32f08137859d44707ec4f086eed9af9b9ee91a87.

## Acknowledgements

The authors thank the patients who volunteered for this study and the staff at Imperial College Healthcare NHS Trust (the Imperial College Healthcare NHS Trust renal COVID-19 group and dialysis staff): Appelbe M, Ashby DR, Brown EA, Cairns T, Charif R, Condon M, Corbett RW, Duncan N, Edwards C, Frankel A, Griffith M, Harris S, Hill P, Kousios A, Levy JB, Loucaidou M, Lightstone L, Liu L, Lucisano G, Lynch K, Mclean A, Moabi D, Muthusamy A, Nevin M, Palmer A, Parsons D, Prout V, Salisbury E, Smith C, Tam F, Tanna A, Tansey K, Tomlinson J, Webster P.

We also acknowledge the efforts of renal specialist doctors in training for assistance with recruiting patients to this study.

We acknowledge support from UKRI/NIHR through the UK Coronavirus Immunology Consortium (UK-CIC) and the National Institute for Health Research (NIHR) Biomedical Research Centre based at Imperial College Healthcare NHS Trust and Imperial College London. The views expressed are those of the author(s) and not necessarily those of the NHS, the NIHR or the Department of Health.

We thank:

Hari and Rachna Murgai and Milan and Rishi Khosla for generous support with sample transport.

Dr Kerry Rostron for exceptional support with laboratory facilities in challenging circumstances and the Department administrators for their help.

Dr Brian Tom and Dr Jessica Barrett (MRC Biostatistics Unit, University of Cambridge) for statistical advice.

Prof Sir John Savill (Melbourne Academic Centre for Health) for comments on the manuscript.

## Additional information

### Competing interests

Stephen P McAdoo: reports personal fees from Celltrion, Rigel, GSK and Cello, outside the submitted work. James E Peters: has received travel and accommodation expenses and hospitality from Olink to speak at Olink-sponsored academic meetings. The other authors declare that no competing interests exist.

## Funding

| Funder | Grant reference number | Author |
| --- | --- | --- |
| UK Research and Innovation | COVID-19 Rapid Response Rolling Call (MR/V027638/1) | James Edward Peters |
| Imperial College London | Community Jameel and the Imperial President's Excellence Fund | James Edward Peters |
| UK Research and Innovation | UKRI Innovation Fellowship at Health Data Research UK (MR/S004068/2) | James Edward Peters |
| Wellcome Trust | Wellcome-Beit Prize Clinical Research Career Development Fellowship (206617/A/17/A) | David C Thomas |
| Wellcome Trust | Wellcome Trust Senior Fellow in Clinical Science (212252/Z/18/Z) | Matthew C Pickering |
| Wellcome Trust | Wellcome Trust and Imperial College London Research Fellowship | Nicholas Medjeral-Thomas Eleanor Sandhu |
| Auchi Renal Research Fund | Auchi Clinical Research Fellowship | Candice L Clarke |
| Medical Research Council | MC_UU_00002/13 | Paul DW Kirk |
| The Sidharth Burman endowment | | David C Thomas |

The funders had no role in study design, data collection and interpretation, or the decision to submit the work for publication.

## Author contributions

Jack Gisby, Data curation, Formal analysis, Investigation, Visualization, Methodology, Writing - original draft; Candice L Clarke, Nicholas Medjeral-Thomas, Maria F Prendecki, Data curation, Investigation, Patient recruitment and sample collection, Clinical phenotyping; Talat H Malik, Paige M Mortimer, Norzawani B Buang, Marie Pereira, Frederic Toulza, Ester Fagnano, Marie-Anne Mawhin, Emma E Dutton, Lunnathaya Tapeng, Investigation, Sample processing; Artemis Papadaki, Formal analysis, Investigation, Methodology; Shanice Lewis, Data curation, Investigation, Sample processing, Clinical phenotyping; Arianne C Richard, Visualization, Methodology, Writing - review and editing, Results interpretation; Paul DW Kirk, Methodology, Statistical methodology; Jacques Behmoaras, Writing - review and editing, Results interpretation; Eleanor Sandhu, Investigation, Patient recruitment and sample collection, Clinical phenotyping; Stephen P McAdoo, Investigation, Patient recruitment and sample collection; Matthew C Pickering, Conceptualization, Data curation, Funding acquisition, Investigation, Project administration, Writing - review and editing, Conceived and designed the study, Clinical phenotyping, Results interpretation, Funding acquisition; Marina Botto, Conceptualization, Resources, Funding acquisition, Investigation, Project administration, Writing - review and editing, Conceived and designed the study, Sample processing, Results interpretation, Funding acquisition; Michelle Willicombe, Funding acquisition, Investigation, Project administration, Conceived and designed the study, Patient recruitment and sample collection, Led and coordinated patient recruitment.; David C Thomas, Conceptualization, Funding acquisition, Investigation, Project administration, Writing - review and editing, Conceived and designed the study. Patient recruitment and sample collection. Results interpretation; James E Peters, Conceptualization, Data curation, Supervision, Investigation, Visualization, Methodology, Writing - original draft, Project administration, Conceived and designed the study, Devised the analysis plan and supervised the analysis, Curated clinical phenotyping data, Funding acquisition

## Author ORCIDs

Jack Gisby (iD) https://orcid.org/0000-0003-0511-8123
Marie Pereira (iD) http://orcid.org/0000-0003-0711-3385

Arianne C Richard (iD) https://orcid.org/0000-0002-8708-9997
Maria F Prendecki (iD) https://orcid.org/0000-0001-7048-7457
Marina Botto (iD) https://orcid.org/0000-0002-1458-3791
James E Peters (iD) https://orcid.org/0000-0002-9415-3440

## Ethics

Human subjects: All participants (patients and controls) were recruited from the Imperial College Renal and Transplant Centre and its satellite dialysis units, London, and provided written informed consent prior to participation. Study ethics were reviewed by the UK National Health Service (NHS) Health Research Authority (HRA) and Health and Care Research Wales (HCRW) Research Ethics Committee (reference 20/WA/0123: The impact of COVID-19 on patients with renal disease and immunosuppressed patients). Ethical approval was given.

## Decision letter and Author response

Decision letter https://doi.org/10.7554/eLife.64827.sa1
Author response https://doi.org/10.7554/eLife.64827.sa2

# Additional files

## Supplementary files

- Source data 1. Individual-level plasma proteomic data for subcohort A.
- Source data 2. Individual-level clinical and demographic covariate data for subcohort A.
- Source data 3. Individual-level serum proteomic data for subcohort B.
- Source data 4. Individual-level clinical and demographic covariate data for subcohort B.
- Supplementary file 1. Table legends. (a) Protein annotation. List of the 436 proteins measured. GeneID = gene symbol of the gene encoding the protein (used as the main identifier in the manuscript); UniProt = UniProt ID; Olink Assay Name = protein id used by Olink; Protein Name = full protein name; Panel name = the name of the 92 protein multiplex Olink panel on which the protein was measured. (b) Enrichment of Reactome terms for the entire set of proteins measured. The results of enrichment testing for genes corresponding to all 436 measured proteins against the background of the genome. The analysis was performed against the Reactome pathways using string-db. The list of Reactome terms is ordered by the number of proteins associated with the term. (c) Differential abundance analysis for COVID-19-positive vs -negative ESKD patients in subcohort A and B. Summary statistics for all 436 proteins are shown. Pvalue = nominal p-value from linear mixed model. Adjusted Pvalue = p-values after Benjamini–Hochberg correction. Fold change = estimated fold change from regression coefficient. Proteins are ordered based on results in subcohort A: first by whether they are significant or not (at 5% FDR), then by fold change (from positive to negative). Note the associations are not ordered by p-value so strong associations do not necessarily appear at the top of the table. Significant adjusted p-values are coloured in green and non-significant in grey. Estimated fold changes are coloured in a gradient from red to blue for up or downregulated in COVID-19 +ve versus –ve, respectively. Sample size for subcohort A: n = 256 plasma samples from 55 COVID-19 positive ESKD patients, versus n = 51 ESKD controls (one sample per control patient). Sample size for subcohort B: 52 samples from 55 COVID-19 patients and 11 non-infected patient samples (single time-point). (d) Associations of proteins and COVID-19 severity (subcohort A). Summary statistics for all 436 proteins are shown. Pvalue = nominal p-value from linear mixed model. Adjusted Pvalue = p-values after Benjamini–Hochberg correction. Fold change = estimated fold change from regression coefficient. Proteins are ordered first by whether they are significant or not (at 5% FDR), then by linear gradient (effect size) from positive to negative. Note the associations are not ordered by p-value so strong associations do not necessarily appear at the top of the table. (e) Predictors of clinical course from Random Forests. Importance metrics for each protein for prediction according to a random forest model trained to predict current or future severe/critical disease using the first sample of each patient. Proteins are ordered by mean minimal depth across all trees – this was used as the primary importance metric. (f) Proteomic predictors of fatal COVID-19. Summary statistics from joint models for fatal disease. Results for all 436 proteins are shown. 'Is significant' indicates significance

(green) or not (grey) at 5% FDR. The association coefficient for each protein indicates the direction and magnitude of the estimated log relative risk for death (red indicates higher protein levels increase risk of death, blue the opposite). 95% confidence intervals are plotted. (g) Associations of proteins and clinical laboratory measurements. Clinical variable = clinical lab tests: white cell count, lymphocyte count, neutrophil count, monocyte count, C-reactive protein, ferritin, d-dimer, troponin. (h) Longitudinal proteomic profiling with linear mixed models. Summary statistics from the linear mixed models used to identify proteins with differential temporal trajectories between mild/moderate (n = 28) and severe/critical COVID-19 patients (n = 27). Summary statistics for all 436 proteins are shown. Pvalue = nominal p-value from linear mixed model for the interaction term between time from symptom onset (days) and overall WHO severity (as a binary variable: mild–moderate or severe–critical). Adjusted Pvalue = p-values after Benjamini–Hochberg correction. 'Is significant' indicates significance (green) or not (grey) at 5% FDR. (i) Comparison to other proteomic studies of COVID-19 positive vs negative patients. Proteins that were differentially abundant in COVID-19 +ve vs -ve patients in our data are listed (5% FDR). TRUE indicates that the protein was reported as differentially abundant in the relevant previous proteomic study. The final column summarises whether the association was previously reported in any of the four studies. We have not harmonised significance thresholds between studies: we simply report whether the authors declared the protein significant by the threshold of their study. (j) Comparison to other proteomic studies of COVID-19 severity. Proteins that were associated with severity in our data are listed (5% FDR). TRUE indicates that the protein was reported as associated with severity in the relevant previous proteomic study. The final column summarises whether the association was previously reported in any one or more of the four studies. We have not harmonised significance thresholds between studies: we simply report whether the authors declared the protein significant by the threshold of their study. Results are shown for all 436 proteins against all eight lab measurements. Adjusted p-value = p-value from linear mixed model after Benjamini–Hochberg correction. Gradient indicates effect size and direction. A positive gradient (red) indicates higher concentrations of proteins are associated with higher clinical laboratory measurements. 'Is significant' indicates significance (green) or not (grey) at 5% FDR. Contemporaneous clinical laboratory tests were not available for all plasma samples. The proportion of samples for which contemporaneous lab tests were available were: white cell count 66%, neutrophils 66%, monocytes 66%, lymphocytes 66%, CRP 64%, ferritin 36%, troponin 35%, d-dimer 30%. (k) Per protein correlations between plasma and serum levels derived from the same blood sample in 11 COVID-19 negative ESKD patients. Plasma and serum were taken from 11 non-infected ESKD patients that were measured in both subcohort A (plasma) and B (serum). Pearson's r was calculated for the 11 paired measurements for each protein. Proteins are ordered by r value; this column is coloured from red to blue for positive and negative r values, respectively. 95% confidence intervals are reported. We also report the variance of the NPX levels for each protein in plasma and in serum.

• Transparent reporting form

## Data availability

All data generated during this study are included in the manuscript and supporting files. Underlying source data for all analyses (individual-level proteomic and clinical phenotyping data) are available without restriction as Source Data Files 1-4. In addition, these data have been deposited in the Dryad Digital Repository (https://doi.org/10.5061/dryad.6t1g1jwxj). Code is available in the following GitHub repository: https://github.com/jackgisby/longitudinal_olink_proteomics copy archived at https://archive.softwareheritage.org/swh:1:rev:32f08137859d44707ec4f086eed9af9b9ee91a87/.

The following dataset was generated:

| Author(s) | Year | Dataset title | Dataset URL | Database and Identifier |
|---|---|---|---|---|
| Gisby J, Clarke CL, Medjeral-Thomas N, Malik TH, Papadaki A, Mortimer PM, Buang NB, Lewis S, | 2020 | Longitudinal proteomic profiling of high-risk patients with COVID-19 reveals markers of severity and predictors of fatal disease | https://doi.org/10.5061/dryad.6t1g1jwxj | Dryad Digital Repository, 10.5061/dryad.6t1g1jwxj |

Pereira M,
Toulza F,
Fagnano E,
Mawhin M,
Dutton EE,
Tapeng L,
Kirk P,
Behmoaras J,
Sandhu E,
McAdoo SP,
Prendecki MF,
Pickering MC,
Botto M,
Willicombe W,
Thomas DC,
Peters JE

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
