## [Decision Letter]

**Acceptance summary:**

The authors analyzed blood proteomic profiles from COVID-19 patients with end-stage renal disease (ESRD) and report changes compatible with an aberrant innate immune response. They also identified several novel molecular predictors of death, as well as markers of severe versus non-severe clinical development in high-risk patients with ESRD infected with SARS-CoV2. Overall, the findings are novel and important as they are the first to characterize potential molecular markers that may help to predict the severity of the clinical course of COVID-19 in this high-risk group of patients.

**Decision letter after peer review:**

Thank you for submitting your article "Longitudinal proteomic profiling of high-risk COVID-19 patients reveals markers of severity and predictors of death" for consideration by *eLife*. Your article has been reviewed by three peer reviewers, including Evangelos J Giamarellos-Bourboulis as the Reviewing Editor and Reviewer #1, and the evaluation has been overseen by a Senior Editor.

The reviewers have discussed the reviews with one another and the Reviewing Editor has drafted this decision to help you prepare a revised submission.

1) The authors are trying to do generalizations for all COVID-19 patients. They need to focus on ESRD which is their main aim. This should be reflected on the title that needs to be appropriately revised.

2) The authors claim that ESRD is a major predisposing factor for COVID-19. This is not yet supported by the existing literature whereas many clinicians feel that patients with ESRD have a milder course than the rest of patients with COVID-19. The authors need to provide support of this (i.e. the selected patients come from how many infected and how was their course compared to the rest of the available cohort) and they need to provide evidence if this is an underlying confounding factor which does not allow the clustering of proteins clearly separating ESRD patients with COVID-19 from ESRD without COVID-19.

3a) The authors discuss about longitudinal sampling whereas they analyze all samples together.

3b) I suggest that they provide analysis of the samples on the day of peak of symptoms that defined the severity classification and compare with samples collected before and after that day.

4) The authors need to provide analysis how the haemodialysis per se has impacted their findings. This can be provided through adjustment with the time since last dialysis.

5) The non-ESRD patients with COVID-19 do not seem appropriate since they have lower frequency of diabetes as co-morbidity.

6) The validation cohort needs to be described in more detail. Are these just more samples from the same study?

7) Figure 5: Individual measured data points should be added to these time-series to show how well the data is modeled.

8) Supplementary Figure 4: As this is supposed to be a validation, please add lead loadings from the primary cohort (shown in Figure 2—figure supplement 1). How well do these replicate?

9) Supplementary Figure 8: X-axis units are "Relative Log2 Abundance" – are these identical to NPX values?

10) The technical replication using just two samples is not relevant (too small). Better refer to technical replications done by Olink or in the literature with larger sample sizes. Also, the reported Pearson correlation of 0.999 looks overly optimistic and is likely a result from correlating data over a vast range of concentrations. What is the relative error (e.g. sum over the differences between the NPX values for each protein divided by their sums)?

11) It appears that some of the controls have been measured twice, in serum and plasma. These data should be compared and substantial differences reported. This information may be important if proteins are taken forward for follow-up and to interpret differences between future studies.

12) All predictions and models should be compared to models that use relevant clinical parameters alone as a reference and then proteins + clinical parameters to test how much information is added by the protein(s).

13) Some information about the role of OLINK measured proteins in ESKD should be provided (maybe from the literature).

14) Ethnicity appears to play a role in Covid-19 severity. The authors included ethnicity into the models. It would be important to highlight and discuss proteins that associate with ethnicity in light of selecting the most suitable and unconfounded biomarkers in future studies.

15) The authors do not describe the matching procedure. Moreover, as shown in Supplementary Table 1 controls differ significantly in terms of comorbidities.

16) Second, the authors should describe better the validation cohort used. Was from the same site? Was in parallel recruited and if so, why is it independent.

17) The Discussion section is confusing. It should be fully re-written in more hierarchical way outscoring the main important findings. Emphasis should be given in ESRD and not in differences with all other papers of proteomics in COVID-19.

18) The authors cite a preprint by Huang (medRxiv, 5 Oct 2020) which has a similar design. In the meantime, a more complete evaluation of that same dataset has been posted by Filbin et al. (Filbin et al., 2020). It may be worthwhile to take some of their results into account in the Discussion.

---

## [Author Response]

1) The authors are trying to do generalizations for all COVID-19 patients. They need to focus on ESRD which is their main aim. This should be reflected on the title that needs to be appropriately revised.

We have revised the title to “Longitudinal proteomic profiling of dialysis patients with COVID-19 patients reveals markers of severity and predictors of death".

2) The authors claim that ESRD is a major predisposing factor for COVID-19. This is not yet supported by the existing literature whereas many clinicians feel that patients with ESRD have a milder course than the rest of patients with COVID-19. The authors need to provide support of this (i.e. the selected patients come from how many infected and how was their course compared to the rest of the available cohort) and they need to provide evidence if this is an underlying confounding factor which does not allow the clustering of proteins clearly separating ESRD patients with COVID-19 from ESRD without COVID-19.

The reviewer’s comments that end-stage kidney disease is not an established risk factor for severe or fatal COVID-19, and that ESKD patients have a milder disease course, are not correct. Multiple major epidemiological and registry studies provide unequivocal evidence that chronic kidney disease (CKD), particularly ESKD, is a major risk factor for severe or fatal COVID-19.

i) Data from the UK Renal Registry shows that 7- and 14-day mortality for COVID-19 infected in-centre haemodialysis (HD) patients was 11% and 19%, respectively.

ii) Data from the Scottish Renal Registry estimates 30-day mortality for HD patients following a positive COVID-19 test as 22% (https://beta.isdscotland.org/find-publications-and-data/population-health/covid-19/scottish-renal-registry-covid-19-report/ ). As of 31 May 2020, 28.2% of renal replacement therapy patients who had had a positive COVID-19 test had died.

iii) The OpenSAFELY study (Williamson et al., 2020) examined ~17 million UK primary care records and linked these to the UK national COVID-19 mortality register. This enabled identification of risk factors for fatal COVID-19. Impaired renal function was a strong risk factor for death from COVID-19, and the risk was proportional to the degree of renal impairment. Patients with estimated glomerular filtration rate (eGFR) <30ml/min/1.73m^2^ had a hazard ratio (HR) for death of 3.56 after adjustment for age and sex (dialysis patients typically have eGFR <15). The association of impaired renal function with risk of death from COVID-19 persisted after adjustment for all other demographic and clinical risk factors in multivariable analysis (eGFR <30 conferred HR for death of 2.52 in the fully adjusted model). This suggests that ESKD per se is a risk factor for COVID19 rather than simply a marker of multi-morbidity. ESKD was one of the strongest risk factors for death from COVID-19, with similar risk to organ transplant or haematological malignancy.

iv) The ISARIC study (Docherty et al. BMJ 2020;369:m1985) examined 20,133 hospitalised patient records and identified CKD as the comorbidity with the strongest association with death (estimated HR 1.28 in multivariable analysis). This study analysed CKD as one group, so the HR for ESKD patients specifically is not available.

v) In our local cohort of 1,352 in-centre haemodialysis patients, by the end of our study recruitment period (30 May 2020), 315 patients had tested positive for COVID-19 (of which 11 acquired COVID-19 in hospital). Excluding the 11 patients who acquired COVID-19 hospital, 160 (53%) required hospitalisation. Of the 315 patients, 85 died (76 in hospital, 9 without admission), leading to a 27% mortality rate. This figure is in line with the Scottish registry data.

Clearly, these multiple sources of data show that mortality rate for ESKD patients far exceeds that of the general population, and that the mortality rate of the patient sample on whom our proteomic study was performed is in line with that observed elsewhere.

We understand the second part of the reviewers’ comment to be asking whether any potential selection bias for severe cases has influenced the differential expression analysis of COVID-19 positive vs. negative ESKD patients. As can be seen from the figures above, the subset of patients we sampled is not overrepresented for mortality compared to either our local haemodialysis population or the UK more generally. Moreover, the striking concordance between our results and that the study of Filbin et al. of COVID-19 positive vs. negative patients in a non-renal population (see our response to reviewer comment 18 and new Figure 4—figure supplement 1) provides reassurance that our findings of proteins differentially expressed in COVID-19 are robust.

We have expanded the Discussion to reflect the above points.

3a) The authors discuss about longitudinal sampling whereas they analyze all samples together.

To clarify, we performed 5 distinct major analyses. Some, but not all of these, use all available samples. This is explained below:

1) Differential protein abundance analysis between all case samples and control samples (i.e. COVID-19 negative haemodialysis patients).

As the reviewer implies, for this analysis we leveraged the fact that for our cases we had serial samples which enables greater statistical power. To appropriately account for the non-independence of serial samples, we used linear mixed models.

2) Analysis of severity within cases.

For this analysis, we analysed associations between protein levels and clinical severity *at the time of blood sampling*. This analysis is greatly enhanced by the fact that we have serial protein measurements, and sometimes changing clinical severity levels, for a given individual. Again, to appropriately model non-independence of samples from the same individual we used a linear mixed modelling strategy. By leveraging all the data, including intra-individual changes in proteome and severity, we gain power.

3) Supervised learning to predict severe disease.

This analysis was performed using *only the first sample* for each patient to identify a model that could predict whether a patient either had severe disease or would develop it in the future.

4) Identifying death associated proteins through modelling with joint models.

This analysis uses repeated measurements to identify predictors of death from serial data. A joint model combines a linear mixed model and a Cox proportional hazard model and is the appropriate analytical method when the predictor (i.e. the protein) is itself a dynamic entity that may be impacted by the clinical state.

5) Identifying proteins with distinct longitudinal profiles.

This analysis is the “longitudinal” model and explicitly models the time from first symptoms. This model tests a) whether proteins significantly change over time and b) examines whether there were proteins that change over time differently in individuals with a mild versus severe disease course. The latter was performed statistically by testing for an interaction effect between time and clinical course. We have realised that we omitted to show the results from a) in the previous manuscript and these are now included (see “Changes to the manuscript in response to 3a) and b)” below).

3b) I suggest that they provide analysis of the samples on the day of peak of symptoms that defined the severity classification and compare with samples collected before and after that day.

It is our view that the analysis we performed to be preferable to the approach suggested by the reviewer – although we acknowledge that this may not have been clear from our original description of our analyses. Linear mixed modelling appropriately accounts for the relationship between serial samples from the same individuals, and is a widely used, well-established, and principled approach for longitudinal data analysis (see, for example, Laird, N.M. and Ware, J.H., 1982. Random-effects models for longitudinal data. Biometrics, pp.963-974; Verbeke, G., 1997. Linear mixed models for longitudinal data. In Linear mixed models in practice (pp. 63-153). Springer, New York, NY).

In contrast, the reviewer’s suggested approach would limit longitudinal analysis to just 3 samples (the sample from the day of peak severity, and the sample immediately preceding and following this). Apart from the practical difficulties that this would present (e.g. for patients for whom the WHO severity score is mild throughout, which samples should be selected?), we do not believe that a reliable longitudinal proteomic trajectory can be constructed through analysis of just 3 data points.

Moreover, we note that the very high concordance (81.8%) in the proteins we identified as severity-associated and those reported in the study by Filbin et al. (see reviewer comment 18 and new Figure 4—figure supplement 1), providing independent support that our approach is robust.

Changes to the manuscript in response to 3a) and b:)

We appreciate that we did not articulate clearly enough the multiple types of analyses we performed. We thus have made the following changes in response to the reviewer’s comments:

i) We have revised the text and introduced new subheadings in the Results section to make the distinction between these analyses clearer. We have also revised the Discussion so that the main findings are discussed in a more structured manner (see also response to reviewer comment 17). We have also edited the Materials and methods to improve clarity.

ii) We have made a new figure (Figure 7—figure supplement 1) illustrating how the death-associated proteins detected in Analysis 4 above relate to the associations with clinical markers of disease severity (e.g. CRP, lymphocyte count).

iii) We have added results showing proteins that significantly change over time (new Supplementary file 1H), in addition to proteins that have different temporal trajectories between severe and non-severe cases. We have amended the text of the Results, Discussion and Materials and methods accordingly.

iv) We have made plots (new Figure 9 and Figure 9—figure supplement 1) showing the temporal trajectories for *all* proteins that display a significantly different longitudinal profile (5% FDR) between patients with a mild versus severe clinical course, since the shape of trajectories was not evident from the p-values alone.

4) The authors need to provide analysis how the haemodialysis per se has impacted their findings. This can be provided through adjustment with the time since last dialysis.

We agree with the reviewer that it is possible that haemodialysis might impact the plasma proteome. To reduce the potential confounding effects of haemodialysis, we designed the study such that all samples were taken immediately before haemodialysis. For the large majority (86.6%) of samples, last haemodialysis was between 48 and 72 hours prior to blood draw. This consistency in timing of blood sampling reduces the potential for impact of this issue.

To evaluate whether time from last haemodialysis might have nevertheless impacted our results, we performed the analysis suggested by the reviewer, including time from last haemodialysis as a covariate along with age, sex and ethnicity in the regression models. Our results were not substantially changed by this. In the differential abundance analysis of COVID-19 positive vs. negative patients, the correlation between log fold changes estimated with and without inclusion of time from last dialysis as a covariate was >0.99 (Figure 3—figure supplement 4A). Minus log10 p-values were also very highly correlated (r >0.99) (Figure 3—figure supplement 4B). Similarly, in the analysis of severity, the correlation of estimated effect sizes and of -log10 p-values were both > 0.99 (Figure 3—figure supplement 4C-D).

We have amended Supplementary file 1C (COVID-19 + vs. – analysis) and Supplementary file 1D (severity analysis) to show for all proteins the results from a) both the model using only age, sex and ethnicity as covariates, and b) the extended model that also includes time for last dialysis (see under the column header “sensitivity analyses with additional covariates”).

We have added this description of the analysis and the results in the Results and Materials and methods.

5) The non-ESRD patients with COVID-19 do not seem appropriate since they have lower frequency of diabetes as co-morbidity.16) Second, the authors should describe better the validation cohort used. Was from the same site? Was in parallel recruited and if so, why is it independent.

We address reviewers’ points 5 and 16 together due to their similarity.

We matched COVID-19 positive and negative ESKD patients as closely as possible in terms of age, sex and ethnicity. We believe that this close demographic matching and the fact that our controls were also haemodialysis patients represents an advance on previous studies. Unfortunately, exact matching of all comorbidities was not feasible, and this issue also applies to other published COVID19 proteomic studies (where hospitalised COVID-19 patients with a high prevalence of comorbidity have often been compared to healthy controls who, by definition, do not have comorbidities). However, we agree with the reviewers that it is important to check whether the lower prevalence of diabetes in our controls might be a potential confounder.

To address this, we have repeated the differential abundance analysis between COVID-19 positive and COVID-19 negative patients adjusting for diabetes status (i.e. adding diabetes as a covariate along with age, sex and ethnicity). Comparison of effect size estimates and -log10 p-values revealed highly consistent results with the model that used only age, sex and ethnicity (Figure 3—figure supplement 3A-B). We now show the results from the extended model under the column header “sensitivity analyses with additional covariates” in Supplementary file 1C.

In addition to the differences in diabetes prevalence, there were also other differences in the underlying cause of ESKD in cases compared controls. We therefore performed a further sensitivity analysis adjusting for underlying cause of renal failure. This did not make any meaningful difference to our results (Figure 3—figure supplement 3C-D). ESKD is characterised pathologically by small shrunken kidneys, loss of glomeruli and scarring, irrespective of the original cause of the renal injury. Thus, even though there maybe disparate causes, ESKD is a more clinically, biochemically and pathologically homogeneous entity than the renal diseases that lead to it. This may explain the lack of impact of this covariate on the plasma proteome.

We have made changes to the text detailing these sensitivity analyses in the Results, Discussion and Materials and methods.

6) The validation cohort needs to be described in more detail. Are these just more samples from the same study?15) The authors do not describe the matching procedure. Moreover, as shown in Supplementary Table 1 controls differ significantly in terms of comorbidities.

Given the similarities in comments 6 and 15 we have responded to these points together.

The two groups of COVID-19 patients were collected from the same site but there was a small temporal difference in collection and a difference in sample types (plasma versus serum).

To put the collections into context, patients were recruited during a highly challenging period for both clinical care and research during the first UK national lockdown. We initiated patient recruitment on 30^th^ March 2020. At this time, we had very limited access to laboratory facilities and were only able to process serum. Once we had the relevant approvals to open our research laboratory facilities, we switched to collecting plasma (from 8^th^ April 2020). The reason for the switch was that a) although the Olink platform can be used for both serum and plasma, it is optimised for the latter, and b) given the high incidence of micro- and macrovascular thrombosis, we hypothesised that it might be important to measure coagulation factors (which can only be measured in plasma).

The initial sample collection (serum) was considerably smaller than the subsequent collection (plasma). In addition, for the majority of patients in the initial collection, we only had a single timepoint, thus precluding longitudinal analyses. Therefore, we used the larger subcohort as the discovery cohort, and the smaller cohort to provide validation for the case-control analysis.

In terms of clinical characteristics, the validation cohort had a higher proportion of hospitalised and severely ill patients. This was because PCR COVID-19 testing in the UK at that time was largely limited to hospitalised patients. By the time we recruited the discovery cohort, the capacity to perform COVID PCR testing had been expanded and was available for patients attending outpatient haemodialysis. Consequently, we were able to identify and recruit patients with milder disease as well as hospitalised patients. Therefore, the discovery cohort included a broader spread of illness severity.

We used the term “independent” to indicate that the validation cohort was a set of non-overlapping patients from the discovery cohort. However, we can now appreciate that this might be misleading and lead a reader to interpret this as patients from a different centre.

Actions:

– We have removed the term “independent cohort” from the revised manuscript.

– To improve clarity, we now refer to the two patient groups as subcohort A (the larger collection that we previously called the discovery cohort) and subcohort B (what was previously called the validation cohort).

– We have added a new table (Table 2) with characteristics of subcohort B.

7) Figure 5: Individual measured data points should be added to these time-series to show how well the data is modeled.

We now provide these in Figure 9—figure supplement 2. Since the large number of data points makes the modelled trajectories difficult to visualise, we show the modelled trajectories without individual data points in the main display item (Figure 9).

8) Supplementary Figure 4: As this is supposed to be a validation, please add lead loadings from the primary cohort (shown in Figure 2—figure supplement 1). How well do these replicate?

We thank the reviewer for this excellent suggestion. To best demonstrate how the patterns in the PCA for subcohort A compare to those in subcohort B, we have done the following:

We present the PCA on the primary cohort and then project the data from subcohort B onto the same PCA coordinates (so that the loadings are orientated identically) (new Figure 2). This reveals that the axis along which cases and controls are separated in subcohort A clearly divides cases and controls in subcohort B (despite the differences in blood materials used, i.e. plasma vs. serum).

We have amended the text as follows:

Results:

“We used the smaller subcohort B (n=52 serum samples from 46 patients with COVID-19; Materials and methods) for validation. […] This revealed clearer separation of infected and non-infected patients than in subcohort A (Figure 2C-D), perhaps reflecting the higher proportion of hospitalised patients (41 of 46 patients) in subcohort B (Table 2).”

Materials and methods:

“Principal Components Analysis

Singular value decomposition was used to perform PCA on the proteomic data from subcohort A (plasma samples). […] This enabled projection of subcohort B data into the PCA space of subcohort A.”

In addition, we have revised the presentation of the PCA analysis to improve clarity as described below:

i) We now show plots for both PC1 vs. PC2 and PC1 vs. PC3 for each subcohort (new Figure 2).

ii) In Figure 2 we show the samples coloured by case control status, and in Figure 2—figure supplement 1 we show samples coloured by severity.

9) Supplementary Figure 8: X-axis units are "Relative Log2 Abundance" – are these identical to NPX values?

Yes these are NPX values. We were trying to make the plot more accessible to readers not familiar with the Olink platform and used the term “relative” to indicate that the measurements were not in absolute units (e.g. ng/L). However, we can see that this may cause confusion.

We have now changed to the axis units to “NPX (log2 protein abundance)”, and added the definition of NPX in the legend. We have also made this change for all relevant figures (new Figure 5, Figure 6C, Figure 9, Figure 9—figure supplement 1 and Figure 9—figure supplement 2).

10) The technical replication using just two samples is not relevant (too small). Better refer to technical replications done by Olink or in the literature with larger sample sizes. Also, the reported Pearson correlation of 0.999 looks overly optimistic and is likely a result from correlating data over a vast range of concentrations. What is the relative error (e.g. sum over the differences between the NPX values for each protein divided by their sums)?

We accept the reviewers’ point that technical replication using just two samples is too small for reliable inference and we have removed this from the manuscript.

11) It appears that some of the controls have been measured twice, in serum and plasma. These data should be compared and substantial differences reported. This information may be important if proteins are taken forward for follow-up and to interpret differences between future studies.

For the 11 control samples from which we had both serum and plasma, we now present the correlation between plasma and serum levels for each protein in Supplementary file 1K.

We have made the following changes to the text to reflect this:

“For 11 ESKD controls, we had contemporaneous plasma and serum samples. To assess the comparability of these two matrices, we calculated the Pearson’s correlation coefficient between the assays for each protein (Supplementary file 1K). […] Caution should be made in extrapolating these findings to the context of active infection where protein dynamic ranges may be different.”

12) All predictions and models should be compared to models that use relevant clinical parameters alone as a reference and then proteins + clinical parameters to test how much information is added by the protein(s).

We have performed the analysis suggested by the reviewer. We also made a minor modification to our method for estimating model accuracy: we now use 4-fold cross-validation and increased the number of iterations to 100 to ensure stability of the accuracy estimate.

Using the Random Forests supervised learning approach, the accuracy of the models are as follows:

– Using only the clinical data (demographics, comorbidities and clinical laboratory tests): 66%.

– Using only the protein data: 71%.

– Using clinical + protein data 71%.

If our primary goal was to develop a clinically useful test based on Olink plasma proteomics to predict severe disease, then developing a predictive model that combined standard clinical parameters and proteins would be most logical. However, we do not believe there is any realistic prospect of Olink proteomics being applied outside the research context in this pandemic. Our goal in performing the supervised learning was not to develop a predictive tool per se, but to use the predictive model to better understand the biology of severe COVID-19 by identifying proteins that the algorithm selected as important predictors. Interrogation of the model to this end is best done for the analysis that used only the proteomic data.

We have made changes to the text to reflect these new analyses (Results).

13) Some information about the role of OLINK measured proteins in ESKD should be provided (maybe from the literature).

We have added this to the Discussion:

“ESKD is itself likely to significantly impact the plasma proteome. Previous cross-sectional studies have shown that the levels of many circulating proteins have an inverse relationship with eGFR [Naseeb et al., 2015; Christensson et al., 2018]. […] Regardless, this observation of widespread changes in the blood proteome of kidney disease patients emphasises the importance of using COVID-19 ESKD patients rather than healthy individuals as our control group.”

14) Ethnicity appears to play a role in Covid-19 severity. The authors included ethnicity into the models. It would be important to highlight and discuss proteins that associate with ethnicity in light of selecting the most suitable and unconfounded biomarkers in future studies.

We have added an analysis of protein associations with ethnicity.

Our view is that this is best done separately for COVID-19 negative and positive ESKD patients, as in the latter proteins associated with severity may be confounded with ethnicity.

We therefore adopted the following approach:

i) In COVID-19 negative ESKD patients (where there was only 1 sample per patient) we fit the following linear regression model (R notation):

NPX ~ ethnicity + age + sex

This revealed no proteins associated with ethnicity at 5% FDR.

ii) In COVID-19 positive ESKD patients (where there were serial samples from the same patient) we included fit the following linear mixed model (again R notation), with a term to adjust for the potential confounding effect of severity:

NPX ~ ethnicity + age + sex + severity + (1 | individual)

This revealed a single protein (LY75) associated with ethnicity at 5% FDR in subcohort A. In subcohort B no proteins were significant at 5% FDR, although LY75 did have nominal P <0.05. In summary, we did not find evidence for substantial ethnicity effects on the proteins we measured. However, we think these results should be interpreted cautiously as the number of individuals in any one ethnic group were modest and consequently power was limited. We do not believe we can make any strong conclusions on this matter from our data and that larger multi-ethnic proteomic studies are needed to properly address this important question.

We have revised the text to reflect the above (Results and Materials and methods).

17) The Discussion section is confusing. It should be fully re-written in more hierarchical way outscoring the main important findings. Emphasis should be given in ESRD and not in differences with all other papers of proteomics in COVID-19.

We thank the reviewer for this helpful suggestion. We have fully re-written the Discussion with a structure corresponding to the major analyses.

We believe that some comparison with other COVID-19 proteomic studies is important to understand whether our findings are applicable only in the context of ESKD or have more general relevance, but we agree with the reviewer’s suggestion to reduce the emphasis on this. We have revised the text accordingly and we now focus on the paper by Filbin et al. as suggested in reviewer comment 18 below. The interested reader can still find comparisons to other studies in Supplementary file 1I (comparison with other COVID-19 positive vs. negative analyses) and in Supplementary file 1J (comparison with other analyses testing association with severity).

18) The authors cite a preprint by Huang (medRxiv, 5 Oct 2020) which has a similar design. In the meantime, a more complete evaluation of that same dataset has been posted by Filbin et al. (Filbin et al., 2020). It may be worthwhile to take some of their results into account in the Discussion.

We now present a comparison of our data with the analysis of Filbin et al. Despite the clinical differences in the populations studied and the differences in Olink proteomics platform used, we observed strikingly similar results. We compared effect sizes from our differential abundance analysis of COVID-19 +ve and COVID-19 -ve ESKD patients to those of Filbin et al. (COVID-19 +ve respiratory failure vs. COVID-19 -ve respiratory failure). This revealed a strong correlation, r = 0.688, (new figure 4—figure supplement 1). Moreover, we observed very high concordance (81.8%) in proteins associated with clinical severity between the two studies. This indicates that similarities in the COVID-19 plasma proteomic signature across clinical contexts and that our findings have relevance beyond the narrow context of COVID-19 in ESKD patients

We have restructured the Results so there is a subsection on comparisons with previous studies, focussing on the report of Filbin et al. (subsection “Comparisons to other proteomic studies in COVID-19”).

We have also revised the Discussion to reflect this.